# FEEDBACK FAVORS THE GENERALIZATION OF NEURAL ODES

**Jindou Jia**[1,2][*]**, Zihan Yang**[1][*]**, Meng Wang**[1]**, Kexin Guo**[1,2][†]
**Jianfei Yang**[3]**, Xiang Yu**[1,2][†]**, Lei Guo**[1,2]
[1]Beihang University   [2]Hangzhou Innovation Institute of Beihang University
[3]Nanyang Technological University

## ABSTRACT

The well-known generalization problem hinders the application of artificial neural networks in continuous-time prediction tasks with varying latent dynamics. In sharp contrast, biological systems can neatly adapt to evolving environments benefiting from real-time feedback mechanisms. Inspired by the feedback philosophy, we present feedback neural networks, showing that a feedback loop can flexibly correct the learned latent dynamics of neural ordinary differential equations (neural ODEs), leading to a prominent generalization improvement. The feedback neural network is a novel two-DOF neural network, which possesses robust performance in unseen scenarios with no loss of accuracy performance on previous tasks. A linear feedback form is presented to correct the learned latent dynamics firstly, with a convergence guarantee. Then, domain randomization is utilized to learn a nonlinear neural feedback form. Finally, extensive tests including trajectory prediction of a real irregular object and model predictive control of a quadrotor with various uncertainties, are implemented, indicating significant improvements over state-of-the-art model-based and learning-based methods. [‡]

## 1 INTRODUCTION

Stemming from residual neural networks (He et al., 2016), neural ordinary differential equation (neural ODE) (Chen et al., 2018) emerges as a novel learning strategy aiming at learning the latent dynamic model of an unknown system. Recently, neural ODEs have been successfully applied to various scenarios, especially continuous-time missions (Liu & Stacey, 2024; Verma et al., 2024; Greydanus et al., 2019; Cranmer et al., 2020). However, like traditional neural networks, the generalization problem limits the application of neural ODEs in real-world applications.

Traditional strategies like model simplification, fit coarsening, data augmentation, and transfer learning have considerably improved the generalization performance of neural networks on unseen tasks (Rohlfs, 2022). However, these strategies usually reduce the accuracy performance on previous tasks, and large-scale training data and network structures are often required to approximate previous accuracy. The objective of this work is to develop a novel network architecture, acquiring the generalization improvement while preserving the accuracy performance.

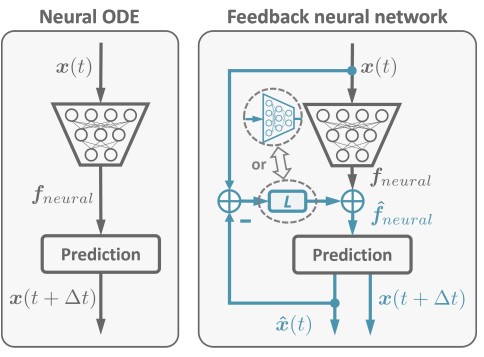

Figure 1: Neural network architectures. *Left:* Neural ODE developed in Chen et al. (2018). *Right:* Proposed feedback neural network.

---

[*]Equal contribution.
[†]Corresponding authors (`{kxguo, xiangyu_buaa}@buaa.edu.cn`).
[‡]Codes are available at `https://sites.google.com/view/feedbacknn`.

Living beings can neatly adapt to unseen environments, even with limited neurons and computing power. One reason can be attributed to the existence of internal feedback (Aoki et al., 2019). Internal feedback has been shown to exist in biological control, perception, and communication systems, handling external disturbances, internal uncertainties, and noises (Sarma et al., 2022; Markov et al., 2021). In neural circuits, feedback inhibition is able to regulate the duration and magnitude of excitatory signals (Luo, 2021). In engineering systems, internal feedback indicates impressive effects across filtering and control tasks, such as *Kalman* filter (Kalman, 1960), *Luenberger* observer (Luenberger, 1966), extended state observer (Guo et al., 2020), and proportional-integral-derivative control (Ang et al., 2005). The effectiveness of feedback lies in its ability to harness real-time deviations between internal predictions/estimations and external measurements to infer dynamical uncertainties. The cognitive corrections are then performed timely. However, existing neural networks rarely incorporate such a real-time feedback mechanism.

In this work, we attempt to enhance the generalization of neural ODEs by incorporating the feedback scheme. The key idea is to correct the learned latent dynamical model of a Neural ODE according to the deviation between measured and predicted states, as illustrated in Figure 1. We introduce two types of feedback: linear form and nonlinear neural form. Unlike previous training methods that compromise accuracy for generalization, the developed feedback neural network is a two-DOF framework that exhibits generalization performance on unseen tasks while maintaining accuracy on previous tasks. The effectiveness of the presented feedback neural network is demonstrated through several intuitional and practical examples, including trajectory prediction of a spiral curve, trajectory prediction of an irregular object and model predictive control (MPC) of a quadrotor.

## 2    NEURAL ODEs AND LEARNING RESIDUES

A significant application of artificial neural networks centers around the prediction task., $\boldsymbol{x}(t) \mapsto \boldsymbol{x}(t + \Delta t)$. Note that $t$ indicates the input $\boldsymbol{x}$ evolves with time. Chen et al. (2018) utilize neural networks to directly learn latent ODEs of target systems, named Neural ODEs. Neural ODEs greatly improve the modeling ability of neural networks, especially for continuous-time dynamic systems (Massaroli et al., 2020), while maintaining a constant memory cost. The ODE describes the instantaneous change of a state $\boldsymbol{x}(t) \in \mathbb{R}^n$

$$\frac{d\boldsymbol{x}(t)}{dt} = \boldsymbol{f}\left(\boldsymbol{x}(t), \boldsymbol{I}(t), t\right) \tag{1}$$

where $\boldsymbol{f}(\cdot) : \mathbb{R}^n \times \mathbb{R}^m \times \mathbb{R} \to \mathbb{R}^n$ represents a latent nonlinear mapping, and $\boldsymbol{I}(t) \in \mathbb{R}^m$ denotes external input. Note that compared with Chen et al. (2018), we further consider $\boldsymbol{I}(t)$ that can extend the ODE to controlled dynamics. The *adjoint sensitive method* is employed in Chen et al. (2018) to train neural ODEs without considering $\boldsymbol{I}(t)$. In Appendix A.1, we provide an alternative training strategy in the presence of $\boldsymbol{I}(t)$, from the view of optimal control.s

Given the ODE (1) and an initial state $\boldsymbol{x}(t)$, future state can be predicted as an initial value problem

$$\boldsymbol{x}(t + \Delta t) = \boldsymbol{x}(t) + \int_t^{t+\Delta t} \boldsymbol{f}\left(\boldsymbol{x}\left(\tau\right), \boldsymbol{I}\left(\tau\right), \tau\right) d\tau. \tag{2}$$

The workflow of neural ODEs is depicted in Figure 1. However, like traditional learning methods, generalization is a major bottleneck for neural ODEs (Marion, 2024). Learning residuals will appear if the network has not been trained properly (e.g., underfitting and overfitting) or the applied scenario has a slightly different latent dynamic model. Take a spiral function as an example (Appendix A.3.1). When a network trained from a given training set (Figure 5 (a)) is transferred to a new case (Figure 5 (b)), the learning performance will dramatically degrade (Figure 5 (d)). Without loss of generality, the learning residual error is formalized as

$$\boldsymbol{f}\left(\boldsymbol{x}(t), \boldsymbol{I}(t), t\right) = \boldsymbol{f}_{neural}\left(\boldsymbol{x}(t), \boldsymbol{I}(t), t, \boldsymbol{\theta}\right) + \Delta\boldsymbol{f}(t) \tag{3}$$

where $\boldsymbol{f}_{neural}\left(\cdot\right) : \mathbb{R}^n \times \mathbb{R}^m \times \mathbb{R} \to \mathbb{R}^n$ represents the learned ODE model parameterized by $\boldsymbol{\theta}$, and $\Delta\boldsymbol{f}(t) \in \mathbb{R}^n$ denotes the unknown learning residual error. In the presence of $\Delta\boldsymbol{f}(t)$, the prediction error of (2) will accumulate over time. The objective of this work is to improve neural ODEs with as few modifications as possible to suppress the effects of $\Delta\boldsymbol{f}(t)$.

## 3 NEURAL ODES WITH A LINEAR FEEDBACK

### 3.1 CORRECTING LATENT DYNAMICS THROUGH FEEDBACK

Even though learned experiences are encoded by neurons in the brain, living organisms can still adeptly handle unexpected internal and external disturbances with the assistance of feedback mechanisms (Aoki et al., 2019; Sarma et al., 2022). The feedback scheme has also proven effective in traditional control systems, facilitating high-performance estimation and control objectives. Examples include *Kalman* filter (Kalman, 1960), *Luenberger* observer (Luenberger, 1966), extended state observer (Guo et al., 2020), and proportional-integral-derivative control (Ang et al., 2005).

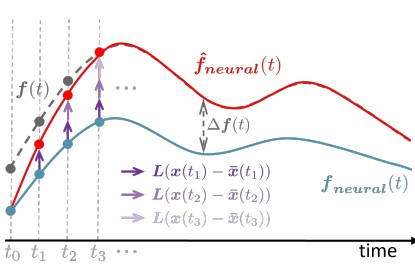

Figure 2: The learned latent dynamics are modified through accumulative evaluation errors to approach the truth one.

We attempt to introduce the feedback scheme into neural ODEs, named feedback neural networks, as shown in Figure 1. Neural ODEs have exploited latent dynamical models $f_{neural}(t)$ of target systems in training set. The key idea of feedback neural networks is to further correct $f_{neural}(t)$ according to state feedback. Denote $t_i$ as the historical evaluation moment satisfying $t_i \leq t$. At current moment $t$, we collect $k + 1$ state measurements $\{x(t_0), x(t_1), \cdots, x(t_k)\}$, in which $t_k = t$. As portrayed in Figure 2, $f_{neural}(t)$ is modified by historical evaluation errors to approach its truth dynamics $f(t)$, i.e.,

$$\hat{f}_{neural}(t) = f_{neural}(t) + \sum_{i=0}^{k} L\left(x\left(t_i\right) - \bar{x}\left(t_i\right)\right) \quad (4)$$

where $L \in \mathbb{R}^{n \times n}$ represents the positive definite matrix and $\bar{x}(t_i) \in \mathbb{R}^n$ represents the predicted state from the last evaluation moment, e.g., an *Euler* integration

$$\bar{x}\left(t_i\right) = x\left(t_{i-1}\right) + T_s \hat{f}_{neural}(t_{i-1}) \quad (5)$$

with the prediction step $T_s \in \mathbb{R}$.

To avoid storing more and more historical measurements over time, define an auxiliary variable

$$\hat{x}\left(t\right) = \bar{x}\left(t\right) - \sum_{i=0}^{k-1} \left(x\left(t_i\right) - \bar{x}\left(t_i\right)\right) \quad (6)$$

where $\hat{x}(t) \in \mathbb{R}^n$ can be regarded as an estimation of $x(t)$. Combining (4) and (6), can lead to

$$\hat{f}_{neural}(t) = f_{neural}(t) + L(x(t) - \hat{x}(t)). \quad (7)$$

From (5) and (6), it can be further rendered that

$$\hat{x}\left(t_k\right) = \hat{x}\left(t_{k-1}\right) + T_s \hat{f}_{neural}(t_{k-1}). \quad (8)$$

By continuating the above *Euler* integration, it can be seen that $\hat{x}(t)$ is the continuous state of the modified dynamics, i.e., $\dot{\hat{x}}(t) = \hat{f}_{neural}(t)$. Finally, $\hat{f}_{neural}(t)$ can be persistently obtained through (7) and (8) recursively, instead of (4) and (5) accumulatively.

### 3.2 CONVERGENCE ANALYSIS

In this part, the convergence property of the feedback neural network is analyzed. The state observation error of the feedback neural network is defined as $\tilde{x}(t) = x(t) - \hat{x}(t)$, and its derivative $\dot{\tilde{x}}(t)$, i.e., the approximated error of latent dynamics is defied as $\tilde{f}(t) = f(t) - \hat{f}_{neural}(t)$. Substitute (1) and (3) into (7), one can obtain the error dynamics

$$\dot{\tilde{x}}(t) = -L\tilde{x}(t) + \Delta f(t). \quad (9)$$

Before proceeding, a reasonable bounded assumption on the learning residual error $\Delta f(t)$ is made.

**Assumption 1.** *There exists an unknown upper bound such that*

$$\|\Delta \boldsymbol{f}(t)\| \leq \gamma \tag{10}$$

*where $\|\cdot\|$ denotes the Euclidean norm and $\gamma \in \mathbb{R}$ is an unknown positive value.*

Note that the above assumption can cover common step disturbances (Figure S12).

**Theorem 1.** *Consider the nonlinear system (1). Under the linear state feedback (7) and the bounded Assumption 1, the state observation error $\tilde{\boldsymbol{x}}(t)$ and its derivative $\dot{\tilde{\boldsymbol{x}}}(t)$ (i.e., $\tilde{\boldsymbol{f}}(t)$) can exponentially converge to bounded sets $\mathcal{B}_1 = \{\tilde{\boldsymbol{x}}(t) \in \mathbb{R}^n : \|\tilde{\boldsymbol{x}}(t)\| \leq \gamma/\lambda_m(\boldsymbol{L})\}$ and $\mathcal{B}_2 = \left\{ \dot{\tilde{\boldsymbol{x}}}(t) \in \mathbb{R}^n : \left\|\dot{\tilde{\boldsymbol{x}}}(t)\right\| \leq \gamma\lambda_M(\boldsymbol{L})/\lambda_m(\boldsymbol{L}) + \gamma \right\}$, respectively, which can be regulated by $\boldsymbol{L}$.*

*Proof.* See Appendix A.2. ∎

### 3.3 MULTI-STEP PREDICTION

With the modified dynamics $\hat{\boldsymbol{f}}(t)$ and current $\boldsymbol{x}(t)$, the next step is to predict $\boldsymbol{x}(t+\Delta t)$ as in (2). By defining $\boldsymbol{z}(t) = \left[\boldsymbol{x}^T(t), \hat{\boldsymbol{x}}^T(t)\right]^T \in \mathbb{R}^{2n}$, from (8), we have $\dot{\boldsymbol{z}}(t) = \left[\boldsymbol{f}^T(t), \hat{\boldsymbol{f}}^T(t)\right]^T$. One intuitional means to obtain $\boldsymbol{z}(t+\Delta t)$ is to solve the ODE problem with modern solvers. However, as shown in Theorem 1, the convergence of $\tilde{\boldsymbol{f}}(t)$ can only be guaranteed as current $t$. In other words, the one-step prediction result by solving the above ODE is accurate, while the error will accumulate in the long-term prediction. In this part, an alternative multi-step prediction strategy is developed to circumvent this problem.

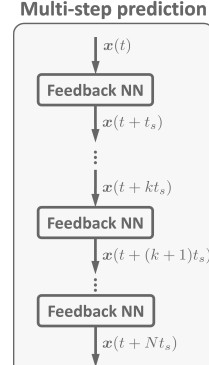

Multi-step prediction

The proposed multi-step prediction strategy is portrayed in Figure 3, which can be regarded as a cascaded form of one-step prediction. The output of each feedback neural network is regarded as the input of the next layer. Take the first two layers as an example. The first-step prediction $\boldsymbol{x}(t + T_s)$ is obtained by $\boldsymbol{x}(t + T_s) = \boldsymbol{x}(t) + \hat{\boldsymbol{f}}(\boldsymbol{x}(t), \hat{\boldsymbol{x}}(t), \theta)T_s$. The second layer with the input of $\boldsymbol{x}(t + T_s)$ will output $\boldsymbol{x}(t + 2T_s)$. In such a framework, the convergence of later layers will not affect the convergence of previous layers. Thus, the prediction error will converge from top to bottom in order.

Figure 3: The multi-step prediction.

Note that the cascaded prediction strategy can amplify the data noise in case of large $\boldsymbol{L}$. A gain decay strategy is designed to alleviate this issue. Denote the feedback gain of $i$-th later as $\boldsymbol{L}_i$, which decays as $i$ increases

$$\boldsymbol{L}_i = \boldsymbol{L} \odot e^{-\beta i} \tag{11}$$

where $\beta$ represents the decay rate. The efficiency of the decay strategy is presented in Figure 5(g). The involvement of the decay factor in the multi-step prediction process significantly enhances the robustness to data noise.

### 3.4 ABLATION STUDY ON OBSERVER GAIN

The adjustment of linear feedback gain $\boldsymbol{L}$ can be separated from the training of neural ODEs, which can increase the flexibility of the structure.

The gain adjustment strategy is intuitional. Theorem 1 indicates that the prediction error will converge to a bounded set as the minimum eigenvalue of feedback gain is positive. And the converged set can shrink with the increase of the minimum eigenvalue. In reality, the amplitude of $\lambda_m(\boldsymbol{L})$ is limited since the feedback $\boldsymbol{x}$ is usually noised. The manual adjustment of $\lambda_m(\boldsymbol{L})$ needs the trade-off between prediction accuracy and noise amplification. Thus, an ablation study on $(\boldsymbol{L})$ to show practical implications of Theorem 1 under Assumption 9 is implemented.

Figure 4 shows the multi-step prediction errors ($N = 50$) with different levels of feedback gains and uncertainties. Two phenomena can be observed from the heatmap. The one is that the prediction error increases with the level of uncertainty. The other is that the prediction error decreases with the gain at the beginning, but due to noise amplification, the prediction error worsens if the gain is set too large.

# 4 NEURAL ODEs WITH A NEURAL FEEDBACK

Section 3 has shown a linear feedback form can promptly improve the adaptability of neural ODEs in unseen scenarios. However, two improvements could be further made. At first, it will be more practical if the gain tuning procedure could be avoided. Moreover, the linear feedback form can be extended to a nonlinear one $\boldsymbol{h}(\boldsymbol{x}(t) - \hat{\boldsymbol{x}}(t)) : \mathbb{R}^n \to \mathbb{R}^n$ to adopt more intricate scenes, as experienced in the control field (Han, 2009).

An effectual solution is to model the feedback part using another neural network, i.e., $\boldsymbol{h}_{neural}(\boldsymbol{x}(t) - \hat{\boldsymbol{x}}(t), \boldsymbol{\xi})$ parameterized by $\boldsymbol{\xi}$. Here we design a separate learning strategy to learn $\boldsymbol{\xi}$. At first, the neural ODE

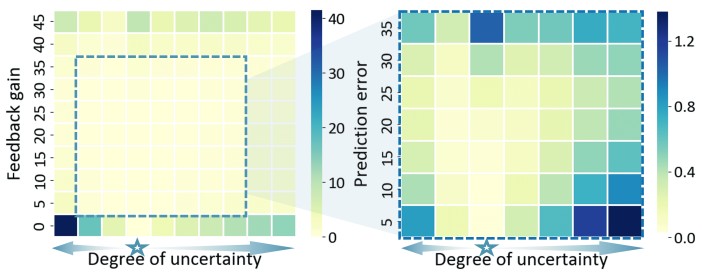

Figure 4: Prediction errors of the spiral curve with different levels of feedback gains and uncertainties to show practical implications of Theorem 1 udner Assumption 9. The *right* image is a partial enlargement of the *left* one. The blue star denotes the case without uncertainty, and the uncertainty increases along both the left and right directions. When the gain is set as 0, the feedback neural network will equal the neural ODE. The related simulation setup is detailed in Appendix A.3.4.

is trained on the nominal task without considering the feedback part. Then the feedback part is trained through domain randomization by freezing the neural ODE. In this way, the obtained feedback neural network is skillfully considered as a two-DOF network. On the one hand, the original neural ODE preserves the accuracy on the previous nominal task. On the other hand, with the aid of feedback, the generalization performance is available in the presence of unknown uncertainties.

## 4.1 DOMAIN RANDOMIZATION

The key idea of domain randomization (Tobin et al., 2017; Peng et al., 2018) is to randomize the system parameters, noises, and perturbations as collecting training data so that the real applied case can be covered as much as possible. Taking the spiral example as an example (Figure 5 (a)), training with domain randomization requires datasets collected under various periods, decay rates, and bias parameters, so that the learned networks are robust to the real case with a certain of uncertainty.

Two shortcomings exist when employing domain randomization. On the one hand, the existing trained network needs to be retrained and the computation burden of training is dramatically increased. On the other hand, the training objective is forced to focus on the average performance among different parameters, such that the prediction ability on the previous nominal task will degraded, as shown in Figure 6 (a). To maintain the previous accuracy performance, larger-scale network designs are often required. In other words, the domain randomization trades precision for robustness. In the proposed learning strategy, the generalization ability is endowed to the feedback loop independently, so that the above shortcomings can be circumvented.

## 4.2 LEARNING A NEURAL FEEDBACK

In this work, we specialize the virtue from domain randomization to the feedback part $\boldsymbol{h}_{neural}(t)$ rather than the previous neural network $\boldsymbol{f}_{neural}(t)$. The training framework is formalized as follows

$$\boldsymbol{\xi}^* = \arg\min_{\boldsymbol{\xi}} \sum_{i=1}^{n_{case}} \sum_{j \in \mathcal{D}_i^{tra}} \left\| \boldsymbol{x}_{i,j}^* - \boldsymbol{x}_{i,j} \right\|$$

$$s.t. \quad \boldsymbol{x}_{i,j} = \boldsymbol{x}_{i,j-1} + T_s \left( \boldsymbol{f}_{neural}(\boldsymbol{x}_{i,j-1}) + \boldsymbol{h}_{neural} \left( \boldsymbol{x}_{i,j-1} - \hat{\boldsymbol{x}}_{i,j-1}, \boldsymbol{\xi} \right) \right) \tag{12}$$

where $n_{case}$ denotes the number of randomized cases, $\mathcal{D}_i^{tra} = \{ \boldsymbol{x}_{i,j-1}, \hat{\boldsymbol{x}}_{i,j-1}, \boldsymbol{x}_{i,j}^* | j = 1, \ldots, m \}$ denotes the training set of the $i$-th case with $m$ samples, $\boldsymbol{x}_{i,j}^*$ denotes the labeled state, and $\boldsymbol{x}_{i,j}$ denotes one-step prediction of state, which is approximated by *Euler* integration method here.

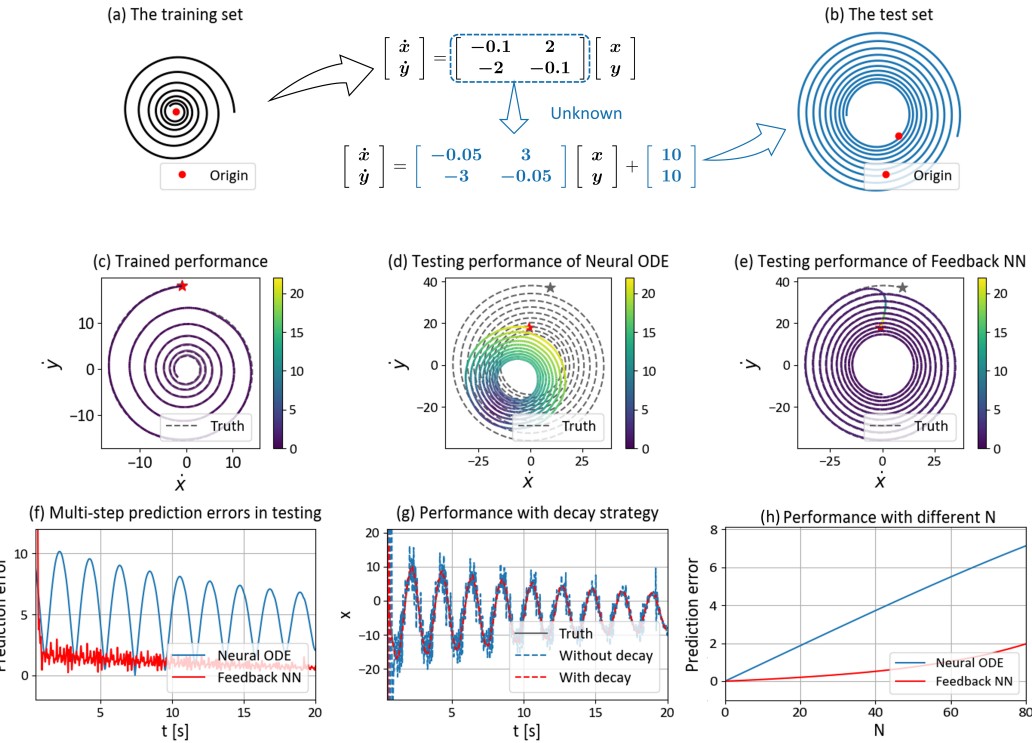

Figure 5: A toy example is presented to intuitively illustrate the developed linear feedback. The mission is to predict the future trajectory of a spiral curve with a given initial state $\{\boldsymbol{x}(t), \boldsymbol{y}(t)\}$. The neural ODE is trained on a given training set (a), yielding an approving learning result (c). Note that the pentagrams denote start points. The trained network is then transferred to a test set (b), which model is significantly different from the training one. With the linear feedback mechanism, the feedback neural network can achieve a better approximated accuracy of the change rate (e), in comparison with the neural ODE (d). As a result, a smaller multi-step prediction error (f) can be attained by benefiting from the feedback neural network. (g) shows that the noise amplification issue in multi-step prediction can be alleviated by the gain-decay strategy. (h) further presents the prediction results with different prediction steps $N$. $N$ in (f)-(g) is set as 50.

The learning procedure of the feedback part $\boldsymbol{h}_{neural}(t)$ is summarized as Algorithm 1. After training the neural ODE $\boldsymbol{f}_{neural}(t)$ on the nominal task, the parameters of simulation model are randomized to produce $n_{case}$ cases. Subsequently, the feedback neural network is implemented in these cases and the training set $\mathcal{D}_i^{tra}$ of each case is constructed. The training loss is then calculated through (12), which favors the update of parameter $\boldsymbol{\xi}$ by backpropagation. The above steps are repeated until the expected training loss is achieved or the maximum number of iterations was reached.

---

**Algorithm 1** Learning neural feedback through domain randomization

---

**Input:** Randomize parameters to produce $n_{case}$ cases; trained neural ODE $\boldsymbol{f}_{neural}$ on nominal task.
**Result:** Neural feedback $\boldsymbol{h}_{neural}$.

    **Initialize:** Network parameter $\boldsymbol{\xi}$; *Adam* optimizer.
1: **repeat**
2:     Run feedback neural network among $n_{case}$ cases to produce $\hat{\boldsymbol{x}}_{i,j}$;
3:     Construct datasets $\mathcal{D}_i^{tra}$;
4:     Evaluate loss through (12) on randomly selected mini-batch data;
5:     Update $\boldsymbol{\xi}$ by backpropagation;
6: **until convergence**

---

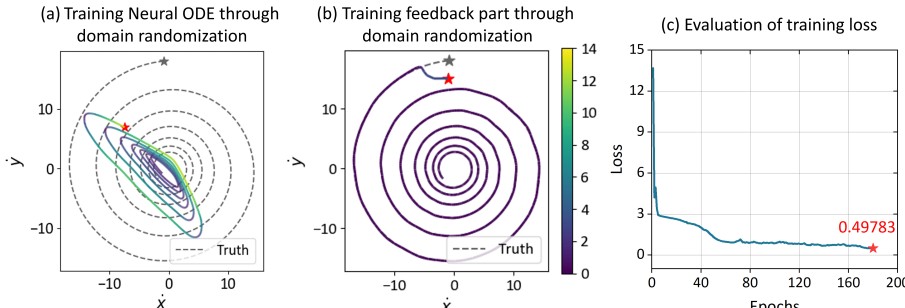

Figure 6: Learning with domain randomization. (a): Train the neural ODE through domain randomization. It can be seen that the learning performance of latent dynamics on the nominal task (Figure 5 (a)) degrades as inducing domain randomization, in comparison with Figure 5 (c). Previous works usually try to scale up neural networks to approach the previous performance. (b): Freeze the neural ODE after training on the nominal task and train the feedback part through domain randomization. The feedback neural network maintains the previous performance on the nominal task. (c) The training loss of the feedback part. Note that the neural ODE employed in (a) and (b) have the same architectures as the one in Figure 5 (c).

For the spiral example, Figure 6 (b) presents the learning performance of the feedback neural network on the nominal task. It can be seen that the feedback neural network can precisely capture the latent dynamics, maintaining the previous accuracy performance of Figure 5 (c). Moreover, the feedback neural network also has the generalization performance on randomized cases, as shown in Appendix Figure S10. Figure 6 (c) further provides the evolution of training loss of the feedback part on the spiral example. More training details are provided in Appendix A.3.3.

## 5 EMPIRICAL STUDY

### 5.1 TRAJECTORY PREDICTION OF AN IRREGULAR OBJECT

Precise trajectory prediction of a free-flying irregular object is a challenging task due to the complicated aerodynamic effects. Previous methods can be mainly classified into model-based scheme (Frese et al., 2001; Müller et al., 2011; Bouffard et al., 2012) and learning-based scheme (Kim et al., 2014; Yu et al., 2021). With historical data, model-based methods aim at accurately fitting the drag coefficient of an analytical drag model, while learning-based ones try to directly learn an acceleration model using specific basis functions. However, the

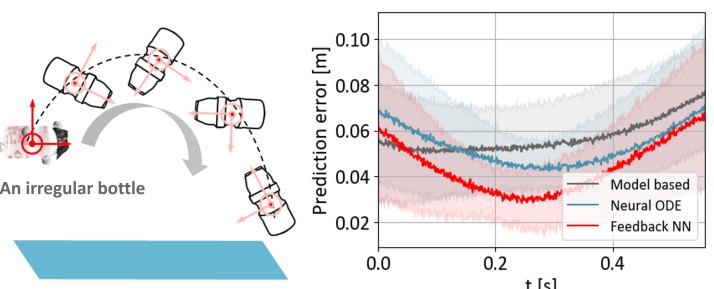

Figure 7: Trajectory prediction results of an irregular bottle. *Left:* The irregular bottle is thrown out by hand and performs an approximate parabolic motion. *Right:* The prediction errors with different methods. The prediction horizon is set as $0.5\ s$. The colored shaded area represents the standard deviations of all 9 test trajectories.

above methods lack of online adaptive ability as employing. Benefiting from the feedback mechanism, our feedback neural network can correct the learned model in real time, leading to a more generalized performance in cases out of training datasets.

We test the effectiveness of the proposed method on an open-source dataset (Jia et al., 2024), in comparison with the model-based method (Frese et al., 2001; Müller et al., 2011; Bouffard et al.,

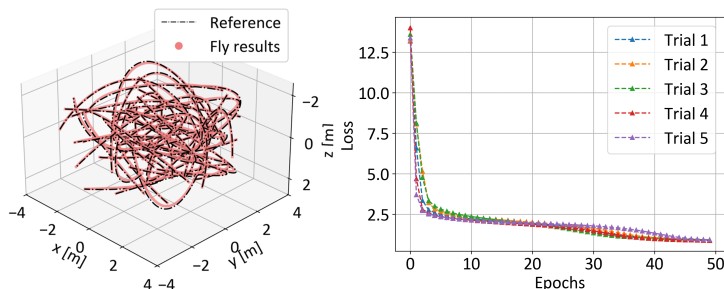

Figure 8: Training sets and convergence procedures. *Left:* Collected trajectories used for training. We first randomly sample positional waypoints in a limited space, followed by optimizing polynomials that connect these waypoints through the minimum snap method (Mellinger & Kumar, 2011). Then the quadrotor with the baseline controller from Jia et al. (2022) is commended to follow planned trajectories, yielding real fly results as the training set. 40 trajectories are collected with the length of 200 discrete nodes each. *Right:* Training curves of 6 random trials. All training trials converged rapidly thanks to stable integration and end-to-end analytic gradients.

2012) and the learning-based method (Chen et al., 2018). The objective of this mission is to accurately predict the object's position after $0.5\ s$, as it is thrown by hand. 21 trajectories are used for training, while 9 trajectories are used for testing. The prediction result is presented in Figure 7. It can be seen that the proposed feedback neural network achieves the best prediction performance. Moreover, the predicted positions and learned latent accelerations of all test trajectories are provided in Figure S2 and Figure S3, respectively. Implementation details are provided in Appendix A.4.

## 5.2 MODEL PREDICTIVE CONTROL OF A QUADROTOR

MPC works in the form of receding-horizon trajectory optimizations with a dynamic model, and then determines the current optimal control input. Approving optimization results highly rely on accurate dynamical models. Befitting from the powerful representation capability of neural networks for complex real-world physics, noticeable works (Torrente et al., 2021; Salzmann et al., 2023; Sukhija et al., 2023) have demonstrated that models incorporating first principles with learning-based components can enhance control performance. However, as the above models are offline-learned within fixed environments, the control performance would degrade under uncertainties in unseen environments.

In this part, the proposed feedback neural network is employed on the quadrotor trajectory tracking scenario concerning model uncertainties and external disturbances, to demonstrate its online adaptive capability. In offline training, a neural ODE is augmented with the nominal dynamics firstly to account for aerodynamic residuals. The augmented model is then integrated with an MPC controller. Note that parameter uncertainties of mass, inertia, and aerodynamic coefficients, and external disturbances are all applied in tests, despite the neural ODE only capture aerodynamic residuals in training. For the feedback neural network, the proposed multi-step prediction strategy is embedded into the model prediction process in MPC. Therefore, the formed feedback-enhanced hybrid model can effectively improve prediction results, further leading to a precise tracking performance. More implementation details refer to Appendix A.5.3.

### 5.2.1 LEARNING AERODYNAMIC EFFECTS

While learning the dynamics, the augmented model requires the participation of external control inputs, i.e., motor thrusts. Earning a quadrotor model augmented with a neural ODE could be tricky with end-to-end learning patterns since the open-loop model are intensively unstable, leading to the diverge of numerical integration. To address this problem, a baseline controller from Jia et al. (2022) is applied to form a stable closed-loop system. The *adjoint sensitive method* is employed in Chen et al. (2018) to train neural ODEs without considering external control inputs. We provide an alternative training strategy concerning external inputs in Appendix A.1, from the view of optimal control. Figure 8 shows training trajectories and convergence procedures. 5 trials of training are

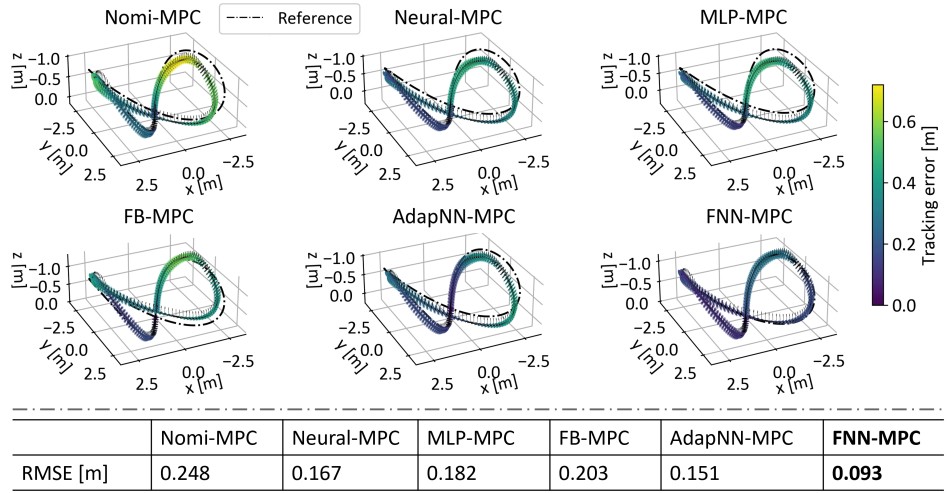

| | Nomi-MPC | Neural-MPC | MLP-MPC | FB-MPC | AdapNN-MPC | **FNN-MPC** |
|---|---|---|---|---|---|---|
| RMSE [m] | 0.248 | 0.167 | 0.182 | 0.203 | 0.151 | **0.093** |

Figure 9: Tracking the *Lissajous* trajectory using MPC with different prediction models.

carried out, each with distinct initial values for network parameters. The trajectory validations are carried out using 3 randomly generated trajectories (Figures S4-S7). More learning details refer to Appendix A.5.2.

### 5.2.2 FLIGHT TESTS

In tests, MPC is implemented with six different models: the nominal model (27), the neural ODE augmented model (Section A.5.2), the feedforward neural network augmented model (Saviolo & Loianno, 2023), the feedback enhanced nominal model, the adaptive neural network augmented model (Cheng et al., 2019) and the proposed feedback neural network, abbreviated as Nomi-MPC, Neural-MPC, MLP-MPC, FB-MPC, AdapNN-MPC, and FNN-MPC, for the sake of simplification. More details of all compared methods refer to Section A.5.4. Moreover, 37.6% mass uncertainty, $[40\%, 40\%, 0]$ inertia uncertainties, $[14.3\%, 14.3\%, 25.0\%]$ drag coefficient uncertainties, and $[0.3, 0.3, 0.3]N$ translational disturbances are applied. The flight results on a *Lissajous* trajectory (out of training set) are presented in Figure 9. The tracking performance is evaluated by root mean square error (RMSE).

It can be seen the Neural-MPC outperforms the Nomi-MPC since intricate aerodynamic effects are captured by the neural ODE. Moreover, the performance of MLP-MPC is relatively unsatisfactory compared with the Neural-MPC. The reason can be attributed to its single-step training manner instead of the multi-step one of the Neural-MPC, leading to a poor multi-step prediction. However, because unseen parameter uncertainties and external disturbances are not involved in the training set, the Neural-MPC still has considerable tracking errors. Due to the adaptive ability of the last layer, AdapNN-MPC can handle a certain level of uncertainty. In contrast, FNN-MPC achieves the best tracking performance. The reason can be attributed to the multi-step prediction of the feedback neural network improves the prediction accuracy subject to multiple uncertainties, as shown in Figure S8.

## 6 RELATED WORK

### 6.1 NEURAL ODES

Most dynamical systems can be described by ODEs. The establishments of ODEs rely on analytical physics laws and expert experiences previously. To avoid such laborious procedures, Chen et al. (2018) propose to approximate ODEs by directly using neural networks, named neural ODEs. The prevalent residual neural networks (He et al., 2016) can be regarded as an *Euler* discretization of neural ODEs Marion et al. (2024). The universal approximation property of neural ODEs has been studied theoretically (Zhang et al., 2020; Teshima et al., 2020; Li et al., 2022), which show the *sup-universality* for $C^2$ diffeomorphisms maps (Teshima et al., 2020) and $L^p$-*universality* for general continuous maps (Li et al., 2022). Marion (2024) further provides the generalization bound (i.e.,

upper bound on the difference between the theoretical and empirical risks) for a wide range of parameterized ODEs.

## 6.2 GENERALIZATION OF NEURAL NETWORKS

In classification tasks, neural network models face the generalization problem across samples, distributions, domains, tasks, modalities, and scopes (Rohlfs, 2022). Plenty of empirical strategies have been developed to improve the generalization of neural networks, such as model simplification, fit coarsening, and data augmentation for sample generalization, identification of causal relationships for distribution generalization, and transfer learning for domain generalization.

Domain randomization (Tobin et al., 2017; Peng et al., 2018) has shown promising effects to improve the generalization for sim-to-real transfer applications, such as drone racing (Kaufmann et al., 2023), quadrupedal locomotion (Choi et al., 2023), and humanoid locomotion (Radosavovic et al., 2024). The key idea is to randomize the system parameters, noises, and perturbations in simulation so that the real-world case can be covered as much as possible. Although the system's robustness can be improved, there are two costs to pay. One is that the computation burden in the training process is dramatically increased. The other is that the training result has a certain of conservativeness since the training performance is an average of different scenarios, instead of a specific case.

## 6.3 REAL-TIME RETRAINING AND ADAPTATION

Recently, online continual learning (Ghunaim et al., 2023) and test-time adaptation (Liang et al., 2024) have emerged as promising solutions to handle unknown test distribution shifts. Online continual learning focuses on the reduction of real-time training load, aiming at generalizing across new tasks while maintaining performance on previous tasks. Test-time adaptation tries to utilize real-time unlabeled data to obtain self-adapted models. For example, an extended *kalman* filter-based adaptation algorithm with a forgetting factor is developed by Abuduweili & Liu (2020) to generalize neural network-based models. Moreover, in order to improve the flexibility of neural networks, the last layer of networks can be regarded as a weighted vector, which can be adjusted adaptively according to real-time state feedback (Cheng et al., 2019; O'Connell et al., 2022; Richards et al., 2023; Saviolo et al., 2024). The training for separating the last layer and front structure can be carried out within a bi-level optimization framework. In such a paradigm, the uncertainty out of training sets is reflected on the last layer of networks, which can be online adjusted in a control-oriented (Richards et al., 2023) or regression-oriented (Cheng et al., 2019; O'Connell et al., 2022) fashion. Patil et al. (2022) further develops real-time weight adaptation laws for all layers of feedforward neural networks, with stability guarantees.

Different from the above retraining or adaptation strategy, the presented method directly corrects the learned latent dynamics of neural ODEs with real-time feedback, yielding a two-DOF network structure. Moreover, the feedback can be learned in a neural form. Integrating adaptive neural ODEs with the developed feedback mechanism may be a valuable research direction (Section A.8).

## 7 CONCLUSION

Inspired by the feedback philosophy in biological and engineering systems, we proposed to incorporate a feedback loop into the neural network structure for the first time, as far as we known. In such a way, the learned latent dynamics can be corrected flexibly according to real-time feedback, leading to better generalization performance in continuous-time missions. The convergence property under a linear feedback form was analyzed. Subsequently, domain randomization was employed to learn a nonlinear neural feedback, resulting in a two-DOF neural network. Finally, applications on trajectory prediction of irregular objects and MPC of robots were shown.

**Limitations.** First, the feedback gain and decay rate for the linear feedback neural network need to be tuned manually. Future work will try to build a bi-level optimization framework to train neural ODE while searching the optimal gains. Such joint optimization manner can also capture the coupled information between feedforward neural ODE and feedback network. Morevoer, the presented nonlinear neural form is preliminarily tested in Section 4. Future work will pursue to exploit its potential in more complex tasks.

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

# A APPENDIX

## A.1 TRAINING NEURAL ODES WITH EXTERNAL INPUTS

Firstly, we formulate the learning problem as an optimization problem:

$$\min_{\boldsymbol{\theta}} \sum_{i=1}^{N-1} l_i(\boldsymbol{x}_i, \boldsymbol{x}_i{}^r, \boldsymbol{\theta}) + l_N(\boldsymbol{x}_N, \boldsymbol{x}_N{}^r) \tag{13}$$

$$s.t. \ \boldsymbol{x}_{i+1} = \boldsymbol{f}_{neural}^d(\boldsymbol{x}_i, \boldsymbol{I}_i, t_i, \boldsymbol{\theta}) \tag{14}$$

where $\boldsymbol{x}_i \in \mathbb{R}^n$ and $\boldsymbol{I}_i \in \mathbb{R}^m$ denotes the model rollout state and the real sample at time $t_i$ respectively, $\boldsymbol{x}_{i+1} = \boldsymbol{f}_{neural}^d(\boldsymbol{x}_i, \boldsymbol{I}_i, t_i, \boldsymbol{\theta})$ refers to the discretized integration of $\boldsymbol{f}_{neural}(\boldsymbol{x}(t), \boldsymbol{I}(t), t, \boldsymbol{\theta})$ with fixed discrete step since real-world state trajectories $\boldsymbol{x}_i{}^r \in \mathbb{R}^n$ are sequentially recorded with fixed timestep based on the onboard working frequency. $l_i(\cdot) \in \mathbb{R}, l_N(\cdot) \in \mathbb{R}$ are defined to quantify the state differences between model rollout $\boldsymbol{x}_i$ and real-world state $\boldsymbol{x}_i{}^r$. In this article, we select the functions in a weighted quadratic form, i.e., $(\boldsymbol{x}_i{}^r - \boldsymbol{x}_i)^\top \boldsymbol{L}_i(\boldsymbol{x}_i{}^r - \boldsymbol{x}_i)$.

By utilizing the optimal control theory and variational method, the first-order optimality conditions of the learning problem could be derived as

$$H = J + \sum_{i=1}^{N-1} \boldsymbol{\lambda}_{i+1}^\top \boldsymbol{f}_{neural}^d(\boldsymbol{x}_i, \boldsymbol{I}_i, t_i, \boldsymbol{\theta}) \tag{15}$$

$$\boldsymbol{x}_{i+1} = \nabla_{\boldsymbol{\lambda}} H = \boldsymbol{f}_{neural}^d(\boldsymbol{x}_i, \boldsymbol{I}_i, t_i, \boldsymbol{\theta}), \ \boldsymbol{x}_1 = \boldsymbol{x}(0) \tag{16}$$

$$\boldsymbol{\lambda}_i = \nabla_{\boldsymbol{x}} H = \nabla_{\boldsymbol{x}} l_i + \left(\frac{\partial \boldsymbol{f}_{neural}^d}{\partial \boldsymbol{x}}\right)^\top \boldsymbol{\lambda}_{i+1}, \ \boldsymbol{\lambda}_N = \frac{\partial l_N}{\partial \boldsymbol{x}_N} \tag{17}$$

$$\frac{\partial H}{\partial \boldsymbol{\theta}} = \sum_{i=1}^{N-1} \nabla_{\boldsymbol{\theta}} l_i + \boldsymbol{\lambda}_{i+1}^\top \nabla_{\boldsymbol{\theta}} \boldsymbol{f}_{neural}^d = 0 \tag{18}$$

where $H \in \mathbb{R}$ stands for the *Hamiltonian* of this problem, $J$ is the objective function in (13). Solving (18) could be done by applying gradient descent on $\boldsymbol{\theta}$. The gradient is analytic and available (summarized in Algorithm 2) by sequentially doing forward rollout (16) of $\boldsymbol{x}$ and backward rollout (17) of $\boldsymbol{\lambda}$, where the latter one is also known as the term *adjoint solve* or *reverse-mode differentiation*.

---

**Algorithm 2** Analytic gradient computation

---

**Input:** Learning objective $l_i(\cdot), l_N(\cdot)$; model $\boldsymbol{f}_{neural}^d$; continuous trajectories $\{\boldsymbol{x}^r(t), \boldsymbol{I}(t), t\}$.
**Result:** Gradient $\partial H / \partial \boldsymbol{\theta}$.
1: $\boldsymbol{x} \leftarrow$ Forward rollout of $\boldsymbol{f}_{neural}^d$ using (16);
2: Compute $\nabla_{\boldsymbol{x}} l_i, \nabla_{\boldsymbol{x}} \boldsymbol{f}_{neural}^d, \nabla_{\boldsymbol{x}_N} l_N, \nabla_{\boldsymbol{\theta}} l_i, \nabla_{\boldsymbol{\theta}} \boldsymbol{f}_{neural}^d$;
3: $\boldsymbol{\lambda} \leftarrow$ Reverse rollout of $\nabla_{\boldsymbol{x}} H$ using (17) ;
4: $\partial H / \partial \boldsymbol{\theta} \leftarrow$ Compute gradient using (18).

---

**Algorithm 3** Training neural ODEs with external inputs

---

**Input:** Learning objective $l_k(\cdot), l_N(\cdot)$; mini-batch size $s$; trajectories $\mathcal{D}^{tra} = \{\boldsymbol{x}^r(t), \boldsymbol{I}(t), t\}$.
**Result:** Neural ODE $\boldsymbol{f}_{neural}^d$.
   **Initialize:** Network parameters $\boldsymbol{\theta}$; slice $\mathcal{D}^{tra}$ into $M$ segments $\{\mathcal{D}_{j=1,\cdots,M}^{tra}\}$ with $s$ length each.
1: **repeat**
2:    **for** $\{\boldsymbol{x}^r{}_{1:s}, \boldsymbol{I}_{1:s}, t_{1:s}\}$ in $\{\mathcal{D}_{j=1,\cdots,M}^{tra}\}$ **do**
3:       Compute analytic gradient $\partial H / \partial \boldsymbol{\theta}$ using Algorithm 2;
4:       Compute learning rate $\alpha$ using *Adam* or other methods;
5:       $\boldsymbol{\theta} \leftarrow \boldsymbol{\theta} - \alpha \cdot \partial H / \partial \boldsymbol{\theta}$;
6:    **end for**
7: **until convergence**

---

The gradient computing only supports for a single continuous state trajectory, and the computational complexity scales linearly with the trajectory length. However, in real-world applications, multiple

trajectory segments with a long horizon might be produced. We introduce mini-batching as well as stochastic optimization methods to deal with the drawback, as summarized in Algorithm 3. The learning rate could be determined using *Adam* or other stochastic gradient descent-related methods.

### A.2 PROOF OF THEOREM 1

#### A.2.1 CONTINUOUS-TIME STABILITY

The proof procedure requires the *Lyapunov* stability analysis arising from the traditional control field (Slotine et al., 1991). At first, define a *Lyapunov* function

$$V(t) = \frac{1}{2}\tilde{\boldsymbol{x}}(t)^T \tilde{\boldsymbol{x}}(t). \tag{19}$$

Differentiate $V(t) \in \mathbb{R}$, yielding

$$
\begin{aligned}
\dot{V}(t) &= \tilde{\boldsymbol{x}}(t)^T \dot{\tilde{\boldsymbol{x}}}(t) \\
&\overset{(a)}{=} \tilde{\boldsymbol{x}}(t)^T \left(-\boldsymbol{L}\tilde{\boldsymbol{x}}(t) + \Delta \boldsymbol{f}(t)\right) \\
&= -\tilde{\boldsymbol{x}}(t)^T \boldsymbol{L}\tilde{\boldsymbol{x}}(t) + \tilde{\boldsymbol{x}}(t)^T \Delta \boldsymbol{f}(t) \\
&\overset{(b)}{\leq} -\frac{\lambda_m(\boldsymbol{L})}{2}\tilde{\boldsymbol{x}}(t)^T \tilde{\boldsymbol{x}}(t) + \frac{1}{2\lambda_m(\boldsymbol{L})}\gamma^2
\end{aligned} \tag{20}
$$

where $(a)$ and $(b)$ are driven by substituting (9) and using *Young*'s inequality $\tilde{x}^T \Delta \boldsymbol{f}(t) \leq \sqrt{\lambda_m(\boldsymbol{L})}\|\tilde{x}\|\frac{\gamma}{\sqrt{\lambda_m(\boldsymbol{L})}} \leq \frac{\lambda_m(\boldsymbol{L})\|\tilde{x}\|^2}{2} + \frac{\gamma^2}{2\lambda_m(\boldsymbol{L})}$, respectively. By combing (19) and (20), it can be rendered that

$$\dot{V}(t) \leq -\lambda_m(\boldsymbol{L})V(t) + \frac{1}{2\lambda_m(\boldsymbol{L})}\gamma^2. \tag{21}$$

By solving the first-order ordinary differential inequality, one can achieve

$$0 \leq V(t) \leq e^{-\lambda_m(\boldsymbol{L})t}\left[V(0) - \delta\right] + \delta \tag{22}$$

with $\delta = \frac{\gamma^2}{[2\lambda_m(\boldsymbol{L})^2]} \in \mathbb{R}$. It can be further implied that

$$\lim_{t \to \infty} \|\tilde{\boldsymbol{x}}(t)\| \leq \frac{\gamma}{\lambda_m(\boldsymbol{L})} \tag{23}$$

which shows that even with learning residuals, the state observation error can converge to a bounded set $\mathcal{B}_1 = \{\tilde{\boldsymbol{x}}(t) \in \mathbb{R}^n : \|\tilde{\boldsymbol{x}}(t)\| \leq \gamma/\lambda_m(\boldsymbol{L})\}$ with the feedback modification. It can be seen that the upper bound can be regulated to arbitrarily small by increasing $\lambda_m(\boldsymbol{L})$.

Finally, from (9), it can be concluded that the derivative of the state observation error can also converge to a bounded set $\mathcal{B}_2 = \left\{\dot{\tilde{\boldsymbol{x}}}(t) \in \mathbb{R}^n : \left\|\dot{\tilde{\boldsymbol{x}}}(t)\right\| \leq \gamma\lambda_M(\boldsymbol{L})/\lambda_m(\boldsymbol{L}) + \gamma\right\}$ with the maximum eigenvalue of feedback gain $\lambda_M(\boldsymbol{L})$.

Figure S1 shows the convergence of the state observation error in the spiral curve example. Related simulational setup is the same as Figure 5. It can be seen that the theoretical bounded set is relatively conservative, as the result of the sufficiency of *Lyapunov* theorem.

As for unbounded learning residuals violating Assumption 1, we think it is still a major challenge in learning fields. It reveals that neural networks have completely lost the representational ability to target uncertainties. The best strategy may be retraining the networks based on fresh datasets, like an online continual learning mission (Ghunaim et al., 2023).

#### A.2.2 DISCRETE-TIME STABILITY

As the developed procedure in (4)-(8) is discrete-time, we further provide the convergence analysis in a discrete-time form.

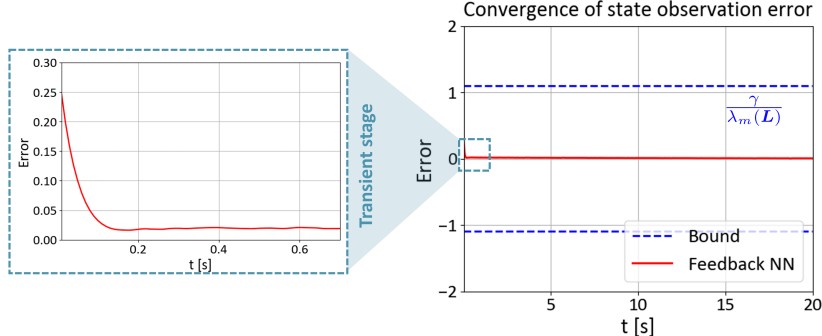

Figure S1: In the spiral curve example, the state observation error can converge to the theoretical bounded set.

Discretizing (2), it can be obtained that

$$\boldsymbol{x}(t_k) = \boldsymbol{x}(t_{k-1}) + T_s \boldsymbol{f}(t_{k-1}). \tag{24}$$

Define state observer error $\tilde{\boldsymbol{x}}(t_k) = \boldsymbol{x}(t_k) - \hat{\boldsymbol{x}}(t_k)$. By making a difference between (24) and (8), one can achieve

$$\begin{aligned}
\tilde{\boldsymbol{x}}(t_k) &= \tilde{\boldsymbol{x}}(t_{k-1}) + T_s(\boldsymbol{f}(t_{k-1}) - \hat{\boldsymbol{f}}_{neural}(t_{k-1})) \\
&\overset{(c)}{=} \tilde{\boldsymbol{x}}(t_{k-1}) + T_s(\boldsymbol{f}_{neural}(t_{k-1}) - \hat{\boldsymbol{f}}_{neural}(t_{k-1}) + \Delta\boldsymbol{f}(t)) \\
&\overset{(d)}{=} (\boldsymbol{I} - T_s\boldsymbol{L})\tilde{\boldsymbol{x}}(t_{k-1}) + T_s\Delta\boldsymbol{f}(t),
\end{aligned} \tag{25}$$

where $(c)$ and $(d)$ are driven by substituting (3) and (7), respectively. With the bounded Assumption 1, if the observer gain $\boldsymbol{L}$ makes $(\boldsymbol{I} - T_s\boldsymbol{L})$ stable, i.e., $\rho(\boldsymbol{I} - T_s\boldsymbol{L}) < 1$, system (25) is input-to-state stable (ISS) (Yan et al., 2023). $\rho(\cdot)$ denotes the spectral radius.

## A.3  IMPLEMENTATION DETAILS OF SPIRAL CASE

### A.3.1  SPIRAL DYNAMICS

The adopted spiral model is formalized as

$$\dot{\boldsymbol{x}}(t) = \begin{bmatrix} -\eta & \omega \\ -\omega & -\eta \end{bmatrix} \boldsymbol{x}(t) + \begin{bmatrix} \varepsilon \\ \varepsilon \end{bmatrix} \tag{26}$$

with period $\omega \in \mathbb{R}$, decay rate $\eta \in \mathbb{R}$, and bias $\varepsilon \in \mathbb{R}$.

In tests, the initial value is set as $\boldsymbol{x}(0) = [9, 0]^T$. For the nominal task, $\omega$, $\eta$, and $\varepsilon$ are set as 2, 0.1, and 0, respectively.

### A.3.2  TRAINING DETAILS OF NEURAL ODE

The adopted MLP for training ODE has 3 layers with 50 hidden units and ReLU activation functions. The training datasets consist of 1000 samples, discretized from $0\,s$ to $10\,s$ with $0.01\,s$ step size. In training, we use *RMSprop* optimizer with the default learning rate of $0.001$. The network is trained with a batch size of 20 for 400 iterations.

### A.3.3  TRAINING DETAILS OF FEEDBACK PART

As for the feedback part, we adopt MLP with 2 hidden layers with 50 hidden units each and ReLU activation functions. The training datasets are collected through domain randomization, with 20 randomized cases, i.e., $\omega = \{0.8 : +0.12 : 3.08\}$, $\eta = \{0.04 : +0.005 : 0.135\}$, $\varepsilon = \{-24 : +2.4 : 21.6\}$. Each case consists of 1000 samples, discretized from $0\,s$ to $20\,s$ with $0.02\,s$ step size. In training, we use *RMSprop* optimizer with the learning rate of $0.01$. The network is trained with a batch size of 100 for 2000 iterations.

### A.3.4 Setup of Gain Adjustment Test

Figure 4 shows the ablation study on linear feedback gain and degree of uncertainty. In this test, feedback gain is selected from $\{0 : +5 : 45\}$ in order, and uncertainties are set as $\omega = \{0.8 : +0.4 : 4.4\}$, $\eta = \{0.04 : +0.02 : 0.22\}$, $\varepsilon = \{-24 : +8 : 96\}$ in order. The prediction step is set as $50$. The prediction results are evaluated using the means of $2\text{-}norm$ prediction errors.

### A.4 Implementation Details of Trajectory Prediction of Irregular Objects

The input state of neural ODE consists of position and velocity. The adopted MLP for training latent ODE has 3 hidden layers with 100 hidden units each and ReLU activation functions. The training datasets consist of 21 trajectories, with 1058 samples each. The step size is $0.001\ s$. In training, we use *Adam* optimizer with the default learning rate of $0.001$. The network is trained with a batch size of 20 for 1000 iterations.

Different from the one-step prediction strategy utilized in Jia et al. (2024) (modeled as dynamical systems concerning attitude), this work predicts future states in a forward-rolling way, learning a more precise result. For the compared drag model-based method, the drag coefficient comes from Jia et al. (2024) fitted by least squares. The prediction error in Figure 7 is evaluated by the $2\text{-}norm$ of position prediction error.

### A.5 Implementation Details of Model Predictive Control of a Quadrotor

#### A.5.1 Quadrotor Preliminaries

A quadrotor dynamics can be defined as a state-space model with a 12-dimensional state vector $\boldsymbol{x} = [\boldsymbol{p}, \boldsymbol{v}, \boldsymbol{\Theta}, \boldsymbol{\omega}]^\top \in \mathbb{R}^{12}$ and a 4-dimensional input vector $\boldsymbol{u} = [T_1, T_2, T_3, T_4]^\top \in \mathbb{R}^4$ of motor thrusts. Two coordinate systems are defined, the earth-fixed frame $\mathcal{E} = \{\boldsymbol{X}_E, \boldsymbol{Y}_E, \boldsymbol{Z}_E\}$ and the body-fixed frame $\mathcal{B} = \{\boldsymbol{X}_B, \boldsymbol{Y}_B, \boldsymbol{Z}_B\}$. The position $\boldsymbol{p} \in \mathbb{R}^3$ and the velocity $\boldsymbol{v} \in \mathbb{R}^3$ are defined in $\mathcal{E}$ while the body rate $\boldsymbol{\omega} \in \mathbb{R}^3$ is defined in $\mathcal{B}$. The relationship between $\mathcal{E}$ and $\mathcal{B}$ is decided by the *Euler* angle $\boldsymbol{\Theta} \in \mathbb{R}^3$. The translational and rotational dynamics can be formalized as

$$
\begin{aligned}
\dot{\boldsymbol{p}} &= \boldsymbol{v}, \quad \dot{\boldsymbol{v}} = \boldsymbol{a} = -\frac{1}{m}\boldsymbol{Z}_B T + g\boldsymbol{Z}_E \\
\dot{\boldsymbol{\Theta}} &= \boldsymbol{W}(\boldsymbol{\Theta})\boldsymbol{\omega}, \quad \boldsymbol{J}\dot{\boldsymbol{\omega}} = -\boldsymbol{\omega} \times (\boldsymbol{J}\boldsymbol{\omega}) + \boldsymbol{\tau} \\
[T, \boldsymbol{\tau}]^\top &= \boldsymbol{C}[T_1, T_2, T_3, T_4]^\top
\end{aligned}
\tag{27}
$$

where $g$ stands for the magnitude of gravitational acceleration, $\boldsymbol{W}(\cdot)$ refers to the rotational mapping matrix of *Euler* angle dynamics and $\boldsymbol{C}$ is the control allocation matrix. We note the nominal dynamics of quadrotor as $\dot{\boldsymbol{x}} = \boldsymbol{f}(\boldsymbol{x}, \boldsymbol{u})$.

Next, differential flatness-based controller (DFBC) (Mellinger & Kumar, 2011) for the quadrotor is introduced, which is adopted here to form a closed-loop system for end-to-end learning that remains stable and differentiable numerical integration. By receiving the flat outputs $\bar{\boldsymbol{\Psi}} = [\boldsymbol{p}, \boldsymbol{v}, \boldsymbol{a}, \boldsymbol{j}] \in \mathbb{R}^{12}$, the positional signal and its higher-order derivatives, as the command signal, DFBC computes the desired motor thrusts for the actuators under the 12-dimensional state feedback. By virtue of the differential flatness property of the quadrotor, one can covert the flat outputs into nominal states $\boldsymbol{x}$ and inputs $\boldsymbol{u}$ using related differential flatness mappings if the yaw motion remains zero. We note this controller as $[\dot{\boldsymbol{z}}, \boldsymbol{u}]^\top = \boldsymbol{\pi}(\boldsymbol{z}, \boldsymbol{x}, \bar{\boldsymbol{\Psi}})$, where $\boldsymbol{z}$ is auxiliary state of controller for the expression integrators and approximated derivatives in the rotational controller.

#### A.5.2 Implemention of Learning Aerodynamics Effects

In training, no external wind and parameter uncertainties exist, and the aerodynamic drag is modeled as $\boldsymbol{R}\boldsymbol{D}\boldsymbol{R}^\top\boldsymbol{v}$ (Faessler et al., 2017), where $\boldsymbol{R}$ refers to the current rotational matrix that maps the frame $\mathcal{B}$ to the frame $\mathcal{E}$, and $\boldsymbol{D} = diag\{[0.6, 0.6, 0.1]\}$ is a coefficient matrix which is fitted by real flight data from Jia et al. (2022).

A neural ODE $\boldsymbol{f}_{neural}$ (with parameters $\boldsymbol{\theta}$) is augmented with the nominal dynamics to capture the aerodynamic effect, i.e., $\dot{\boldsymbol{v}} = \boldsymbol{a} = -\frac{1}{m}\boldsymbol{Z}_B T + g\boldsymbol{Z}_E + \boldsymbol{f}_{neural}(\boldsymbol{v}, \boldsymbol{\Theta}, \boldsymbol{\theta})$. A MLP with 2 hidden layers with 36 hidden units is adopted.

End-to-end learning of $\boldsymbol{f}_{neural}$ could be done using the algorithm 3, but a stable numerical integration is necessary. A closed-loop system of the augmented dynamics using DFBC is employed, noted as $[\dot{\boldsymbol{x}}, \dot{\boldsymbol{z}}]^\top = \boldsymbol{\Phi}([\boldsymbol{x}, \boldsymbol{z}]^\top, \bar{\boldsymbol{\Psi}})$. In the proposed algorithm, $[\dot{\boldsymbol{x}}, \dot{\boldsymbol{z}}]^\top$ turns out to be the new state and $\bar{\boldsymbol{\Psi}}$ becomes the auxiliary input instead of the input of the augmented dynamics $\boldsymbol{u}$.

We generate $40$ $\bar{\boldsymbol{\Psi}}$ trajectories with the discrete nodes of $200$ each for learning by randomly sampling the positional waypoints in a limited space, followed by optimizing polynomials that connect these waypoints, as shown in Figure 8. For validations of the learned neural ODE, we generate another 3 random $\bar{\boldsymbol{\Psi}}$ trajectories $2.5\times$ longer than that used in training, the result illustrated in Figures S4-S7 indicates a good prediction on all 12 states.

### A.5.3 Implementation of MPC with Feedback Neural Networks

MPC works in the form of trajectory optimization (28) with receding-horizon $N$ with a discrete dynamic model $\boldsymbol{f}_d$, to obtain the current optimal control input $\boldsymbol{u}_0$, while maintaining feasibility constraints $\boldsymbol{u}_i \in \mathbb{U}, \boldsymbol{x}_i \in \mathbb{X}$, i.e.,

$$
\begin{aligned}
\min_{\boldsymbol{x}_{1:N}, \boldsymbol{u}_{0:N-1}} \quad & l_N(\boldsymbol{x}_N, \boldsymbol{x}_N^r) + \sum_{i=1}^{N} l_x(\boldsymbol{x}_i, \boldsymbol{x}_i^r) + l_u(\boldsymbol{u}_i, \boldsymbol{u}_i^r) \\
s.t. \quad & \boldsymbol{x}_{i+1} = \boldsymbol{f}_d(\boldsymbol{x}_i, \boldsymbol{u}_i), \ \boldsymbol{x}_0 = \boldsymbol{x}(0) \\
& \boldsymbol{u}_i \in \mathbb{U}, \ \boldsymbol{x}_i \in \mathbb{X}
\end{aligned}
\tag{28}
$$

where the objective functions $l_x(\cdot) \in \mathbb{R}, l_u(\cdot) \in \mathbb{R}, l_N(\cdot) \in \mathbb{R}$ penalize the tracking error between model predicted trajectory $\{\boldsymbol{x}_{1:N}, \boldsymbol{u}_{1:N}\}$ and the up-comming reference trajectory $\{\boldsymbol{x}_{1:N}^r, \boldsymbol{u}_{1:N}^r\}$, where quadratic loss are often adopted. In this application, we make $l_x(\cdot) = l_N(\cdot) = (\boldsymbol{x} - \boldsymbol{x}^r)^\top \boldsymbol{Q} (\boldsymbol{x} - \boldsymbol{x}^r)$, $l_u(\cdot) = (\boldsymbol{u} - \boldsymbol{u}^r)^\top \boldsymbol{R} (\boldsymbol{u} - \boldsymbol{u}^r)$, where $\boldsymbol{Q} = diag\{[\boldsymbol{100}_{3\times1}, \boldsymbol{50}_{6\times1}, \boldsymbol{1}_{3\times1}]\} \in \mathbb{R}^{12\times12}$ and $\boldsymbol{R} = diag\{\boldsymbol{1}_{4\times1}\} \in \mathbb{R}^{4\times4}$. The feasibility constraints $\boldsymbol{u}_i \in \mathbb{U}$ and $\boldsymbol{x}_i \in \mathbb{X}$ are normally designed using box constraints. We make $\boldsymbol{0}_{4\times1} \le \boldsymbol{u} \le \boldsymbol{4}_{4\times1}$ to avoid control saturation and $|\boldsymbol{\Theta}| \le \boldsymbol{\pi/2}_{3\times1}$ to avoid singularities while using *Euler* angle-based attitude representation. The receding horizon length $N$ is set to be 10.

The key idea of using a feedback neural network augmented model is to apply the multi-step prediction mechanism to the model prediction process in MPC. The multi-step prediction algorithm requires the current feedback state $\boldsymbol{x}_0$ and current input $\boldsymbol{u}_1$ to update the sequence of $\hat{\boldsymbol{x}}_{1:N}$. The updated $\hat{\boldsymbol{x}}_{1:N}$ can be directly applied for the next receding horizon optimization. We choose a linear feedback gain of $\boldsymbol{L} = diag\{\boldsymbol{3}_{12\times1}\} \in \mathbb{R}^{12\times12}$ with a decay rate of $0.1$.

### A.5.4 Benchmark comparisons

In the quadrotor example, in order to show the effectiveness of the proposed feedback neural network, five other models are compared: the nominal model (27), the neural ODE augmented model (Section A.5.2), the feedforward neural network augmented model (Saviolo & Loianno, 2023), the feedback enhanced nominal model, and the adaptive neural network augmented model (Cheng et al., 2019), abbreviated as Nomi-MPC, Neural-MPC, MLP-MPC, FB-MPC, and AdapNN-MPC, for the sake of simplification.

The MLP augmented model employs the fully connected neural network to learn aerodynamic drag (Saviolo & Loianno, 2023). The feedback enhanced nominal model refers to the analytic model (27) strengthened by proposed feedback mechanism. The adaptive neural network augmented model (Cheng et al., 2019) uses the feedforward neural network to learn aerodynamic drag in which the last layer is regarded as a weighted vector, being adjusted adaptively according to real-time state feedback. Similar idea is also proposed in O'Connell et al. (2022); Richards et al. (2023); Saviolo et al. (2024). In tests, all learning-based methods have the same hidden layers, and the parameters of the AdapNN-MPC are adjusted for optimal performance.

The training loss on the training set, the validation set, and the test set of MLP augmented model is provided in Figure S13.

### A.5.5 TEST RESULTS

A periodic $3D$ *Lissajous* trajectory is used for comparative tests, where a variety of attitude-velocity combination is exploited. The position trajectory can be written as $\boldsymbol{p}(t) = [r_x sin(2\pi t/T_x), \ r_y sin(2\pi t/T_y), \ h + r_z cos(2\pi t/T_z)]$, where the parameters are $[r_x, r_y, r_z, T_x, T_y, T_z, h] = [3.0, 3.0, 0.5, 6.0, 3.0, 3.0, 0.5]$. Tracking such trajectory requires a conversion of the flat outputs to the nominal 12-dimensional state $\boldsymbol{x}$ of the quadrotor using differential flatness-based mapping.

During trajectory tracking, it could be seen from Figure S8 that the prediction accuracy of latent dynamics at the first step is improved significantly under the multi-step prediction. Although the learning-based model provides more solid results on dynamics prediction than just using the nominal model, with the help of feedback, a convergence property of prediction error can be achieved, leading to a better tracking performance (Figure 9).

### A.6 ABLATION STUDY

Section 3.4 has analyzed the sensitivity of observer gain at different levels of uncertainties. In this part, we further conduct the ablation studies on linear and nonlinear neural feedback units, and decay rate.

### A.6.1 LINEAR FEEDBACK UNIT

We test the performance of correcting the latent dynamics of spiral curves at 12 different levels of learning residuals, with or without linear feedback unit. The parameter uncertainties cover $\Delta\omega = \{-0.72 : +0.12 : 0.6\}$, $\Delta\eta = \{-0.03 : +0.005 : 0.025\}$, $\Delta\varepsilon = \{-14.4 : +2.4 : 12\}$. All compared results are summarized in Figure S9, which indicates the effectiveness of the linear feedback unit.

### A.6.2 NEURAL FEEDBACK UNIT

Similar to the last test, we further test the performance of the neural feedback unit at 12 different levels of learning residuals. All compared results are summarized in Figure S10. It can be found that the developed feedback neural network shows better generalization performance by enabling the neural feedback unit.

It can be seen from Figure S9 and Figure S10, both methods can achieve comparative learning performance. Compared with the linear feedback, no prior gain tunning is required for the neural feedback at the cost of training cost.

### A.6.3 DECAY RATE

Ablation study on decay rate: The performance of the decay rate is examined in the spiral curve example. In tests, the decay rate is set as $\beta = \{0 : +0.01 : 0.06\}$ in sequence, and the multi-step prediction errors (Figure 5(g)) are calculated in RMSE. The test results are shown in the Figure S11. It can be seen that the prediction error decreases with the increase of $\beta$ at the beginning due to noise mitigation. However, as $\beta$ continues to increase, the convergence time becomes slower, leading to a gradual increase in prediction error.

In practice, the tunning of feedback gain and decay rate is very intuitive. They can be increased slowly from a small value until the critical value with the best estimation performance is reached.

### A.7 TRAINING COST

For the training of neural ODEs, two strategies are employed in this work: the adjoint sensitive method developed in Chen et al. (2018) without considering external inputs, and the alternative training method developed in Appendix A.1 concerning external inputs. The adjoint sensitive method is utilized in the spiral curve and irregular object examples, and its computational resource and training time are the same as Chen et al. (2018). The alternative training method concerning external inputs is employed in the quadrotor example to learn residual dynamics. It takes around 30 mins to run 50 epochs on a laptop with 13th Gen Intel(R) Core(TM) i9-13900H. The alternative training

method is derived from the view of optimal control, and its computational resource and training time are comparable to the adjoint sensitive method theoretically.

Two feedback forms are presented. No prior training is required for the linear feedback form, showing its advantage over traditional learning-based generalization methods. Moreover, the linear feedback consists of several analytic equations, consuming almost no computing resources. As for the neural feedback form, due to the optimization problem being non-convex, a satisfactory result usually takes 10 mins to 1 hour of training time on a laptop with Intel(R) Core(TM) Ultra 9 185H 2.30 GHz.

### A.8 COMBINATION WITH ADAPTIVE NEURAL ODE

In this part, we further explore the combination potential of the developed linear feedback neural network with the test-time adaptation (Cheng et al., 2019; O'Connell et al., 2022; Richards et al., 2023; Saviolo et al., 2024). Let $\boldsymbol{f}_{neural}(\boldsymbol{x}(t), \boldsymbol{I}(t), t, \boldsymbol{\theta}) = \boldsymbol{\Xi}(\boldsymbol{x}(t), \boldsymbol{I}(t), t, \boldsymbol{\theta})\boldsymbol{\chi}$, where $\boldsymbol{\Xi}(\cdot) : \mathbb{R}^n \times \mathbb{R}^m \times \mathbb{R} \to \mathbb{R}^n \times \mathbb{R}^l$ represents front layers of neural network, and $\boldsymbol{\chi} \in \mathbb{R}^l$ denotes the weighted vector of the last layer of neural network, which is constant in a test case and can be adjusted adaptively according to real-time state feedback. Integrating with the adaptive scheme, (7) can be adjust to

$$\hat{\boldsymbol{f}}_{neural}(t) = \boldsymbol{\Xi}(t)\hat{\boldsymbol{\chi}} + \boldsymbol{L}(\boldsymbol{x}(t) - \hat{\boldsymbol{x}}(t)), \tag{29}$$

where $\hat{\boldsymbol{\chi}}$ is updated through an adaptive law

$$\dot{\hat{\boldsymbol{\chi}}} = \boldsymbol{\Gamma}\boldsymbol{\Xi}^T(t)\tilde{\boldsymbol{x}}(t) \tag{30}$$

with a positive definite observer gain $\boldsymbol{\Gamma} \in \mathbb{R}^l \times \mathbb{R}^l$.

**Theorem 2.** *Consider the nonlinear system (1). Under the linear state feedback (29), the adaptive law (30), and the bounded Assumption 1, the state observation error $\tilde{\boldsymbol{x}}(t)$ and its derivative $\dot{\tilde{\boldsymbol{x}}}(t)$ (i.e., $\tilde{\boldsymbol{f}}(t)$) can exponentially converge to bounded sets $\mathcal{B}_1 = \{\tilde{\boldsymbol{x}}(t) \in \mathbb{R}^n : \|\tilde{\boldsymbol{x}}(t)\| \leq \gamma/\lambda_m(\boldsymbol{L})\}$ and $\mathcal{B}_2 = \left\{\dot{\tilde{\boldsymbol{x}}}(t) \in \mathbb{R}^n : \left\|\dot{\tilde{\boldsymbol{x}}}(t)\right\| \leq \gamma\lambda_M(\boldsymbol{L})/\lambda_m(\boldsymbol{L}) + \gamma\right\}$, respectively, which can be regulated by $\boldsymbol{L}$.*

*Proof.* Define the estimation error $\tilde{\boldsymbol{\chi}} = \boldsymbol{\chi} - \hat{\boldsymbol{\chi}}$ and a *Lyapunov* function

$$V(t) = \frac{1}{2}\tilde{\boldsymbol{x}}(t)^T\tilde{\boldsymbol{x}}(t) + \frac{1}{2}\tilde{\boldsymbol{\chi}}^T\boldsymbol{\Gamma}^{-1}\tilde{\boldsymbol{\chi}}. \tag{31}$$

Differentiate $V(t) \in \mathbb{R}$, yielding

$$\begin{aligned}
\dot{V}(t) &= \tilde{\boldsymbol{x}}(t)^T(\dot{\boldsymbol{x}}(t) - \dot{\hat{\boldsymbol{x}}}(t)) - \tilde{\boldsymbol{\chi}}^T\boldsymbol{\Gamma}^{-1}\dot{\hat{\boldsymbol{\chi}}} \\
&\overset{(e)}{=} -\tilde{\boldsymbol{x}}(t)^T\boldsymbol{L}\tilde{\boldsymbol{x}}(t) + \tilde{\boldsymbol{x}}(t)^T(\boldsymbol{\Xi}(t)\tilde{\boldsymbol{\chi}} + \Delta\boldsymbol{f}(t)) - \tilde{\boldsymbol{\chi}}^T\boldsymbol{\Xi}^T(t)\tilde{\boldsymbol{x}}(t) \\
&= -\tilde{\boldsymbol{x}}(t)^T\boldsymbol{L}\tilde{\boldsymbol{x}}(t) + \tilde{\boldsymbol{x}}(t)^T\Delta\boldsymbol{f}(t)
\end{aligned} \tag{32}$$

where $(e)$ is driven by substituting (9), (29), and (30). The following proof process is consistent with (20), which is omitted here.

Compared (7), (29) further increases the flexibility of the neural network by inducing the adaptive mechanism. We further test this scheme in the quadrotor example, as shown in Figure S14. Note that the feedback gain is set the same as that of the previous feedback neural network. It can seen that the adaptation-enhanced feedback neural network (abbreviated as AdapFNN) achieves performance comparable to the previous feedback neural network, with a slightly larger RMSE.

We think that the possible reason why the AdapFNN does not bring significant performance improvement is that the last layer of the neural network is not trained analytically. In other words, the uncertainty of the test scenario is not reflected in the last layer. The bilevel training strategy (O'Connell et al., 2022; Richards et al., 2023) may help improve AdapFNN's performance. $\quad\square$

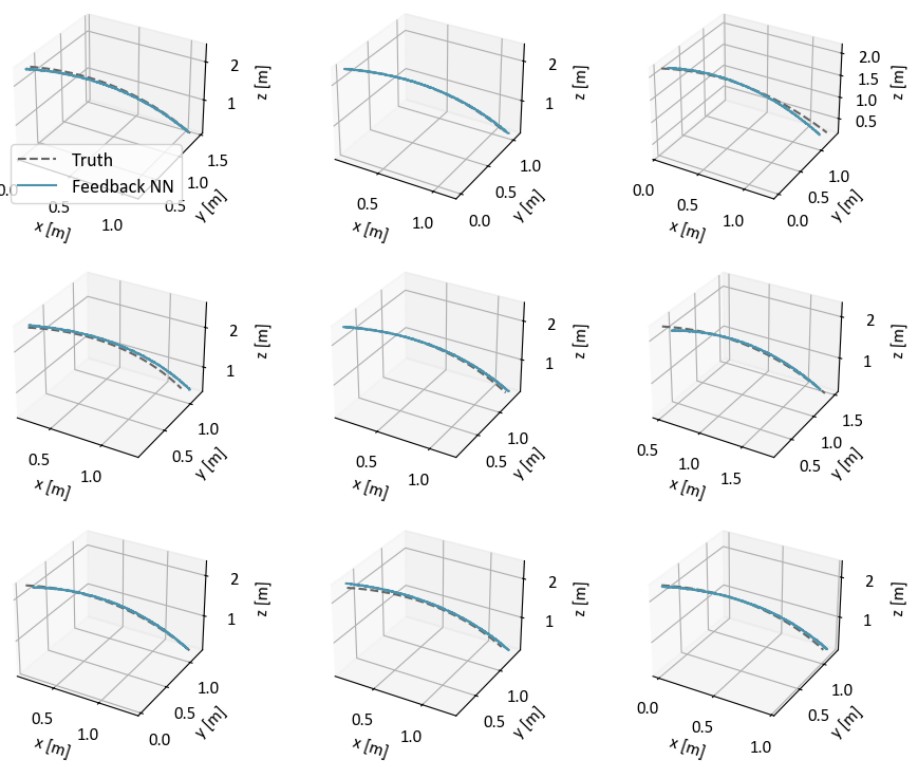

Figure S2: Trajectory prediction results of all 9 test trajectories in Section 5.1. It can be seen that the predicted trajectories almost overlap with the truth ones.

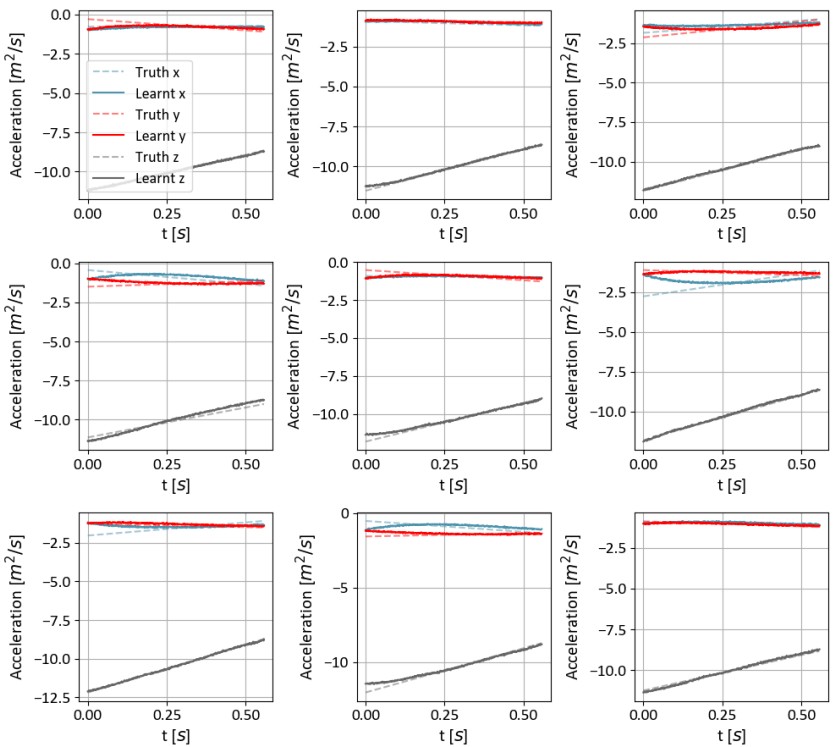

Figure S3: The learning performance of latent accelerations of all 9 test trajectories in Section 5.1. It can be seen that the feedback neural network can accurately capture the latent dynamics of test trajectories out of the training set.

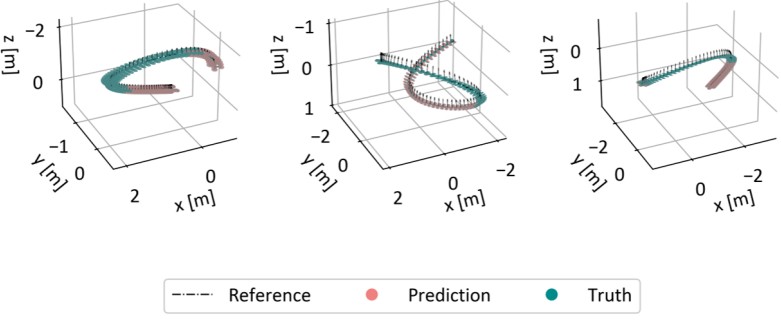

Figure S4: 3 random trajectories generated for validations of the learned neural ODE, named traj-#1, traj-#2, and traj-#3. All trajectories show well-predicted motions on pose and attitude. Detailed results on all 12 states are provided in Figures S5-S7.

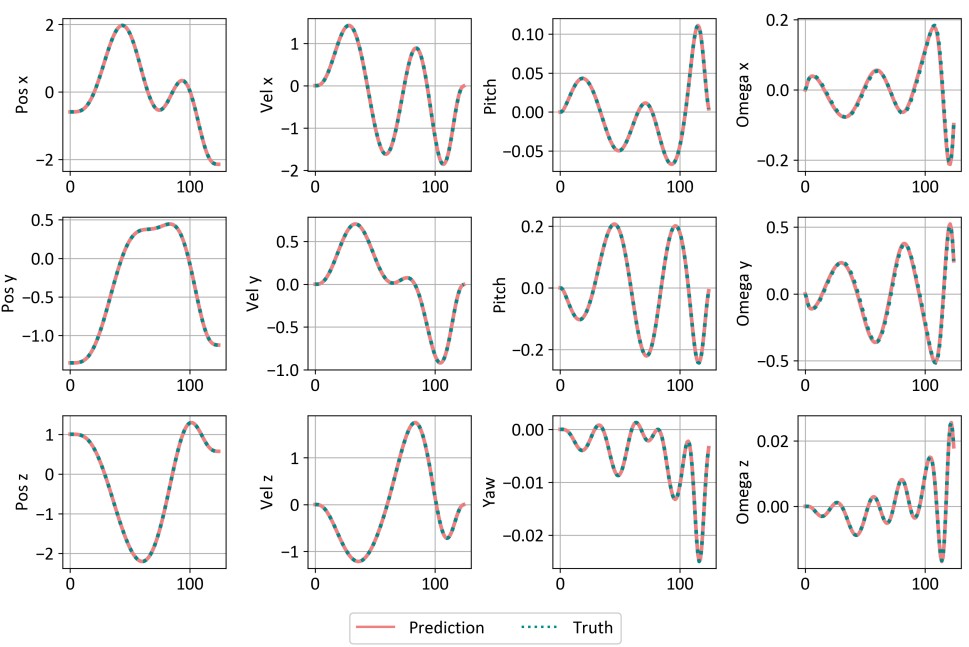

Figure S5: Validation of learned neural ODE. Prediction on all 12 states of traj-#1.

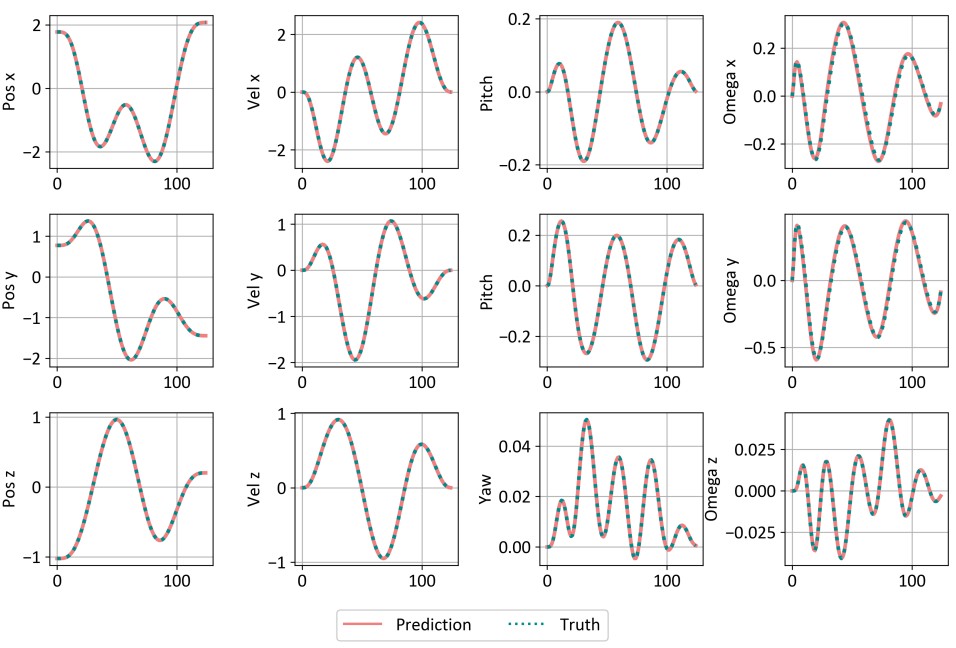

Figure S6: Validation of learned neural ODE. Prediction on all 12 states of traj-#2.

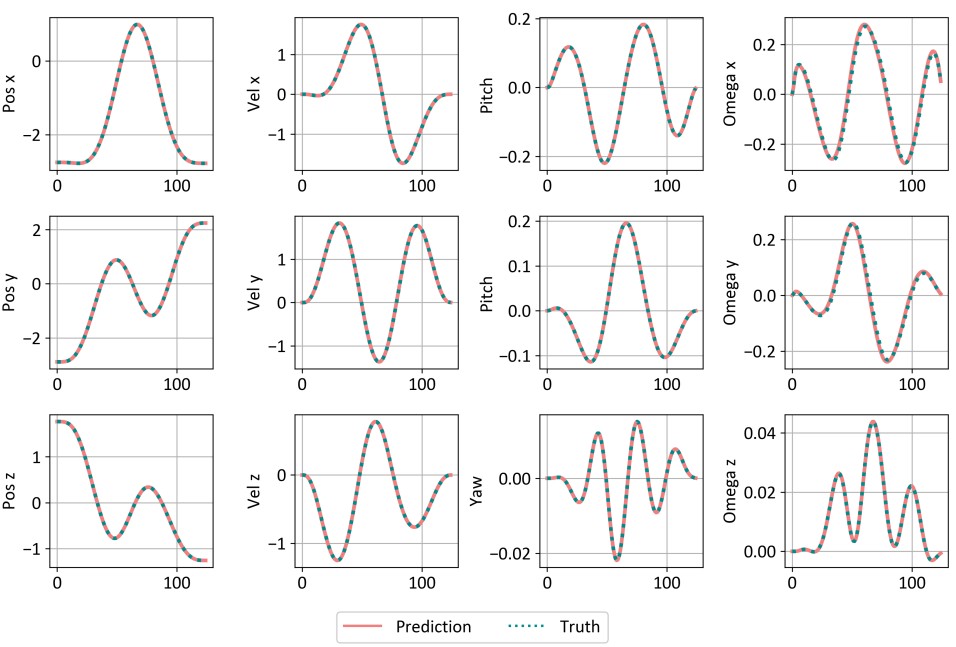

Figure S7: Validation of learned neural ODE. Prediction on all 12 states of traj-#3.

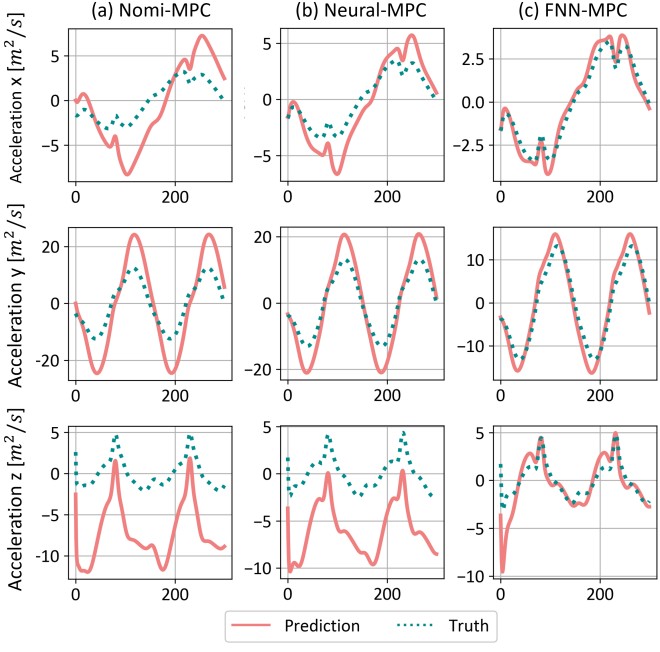

Figure S8: Test on the *Lissajous* trajectory. Prediction on the translational latent dynamics (i.e., acceleration) at the first step using different prediction models. The feedback neural network augmented model achieve the best prediction performance.

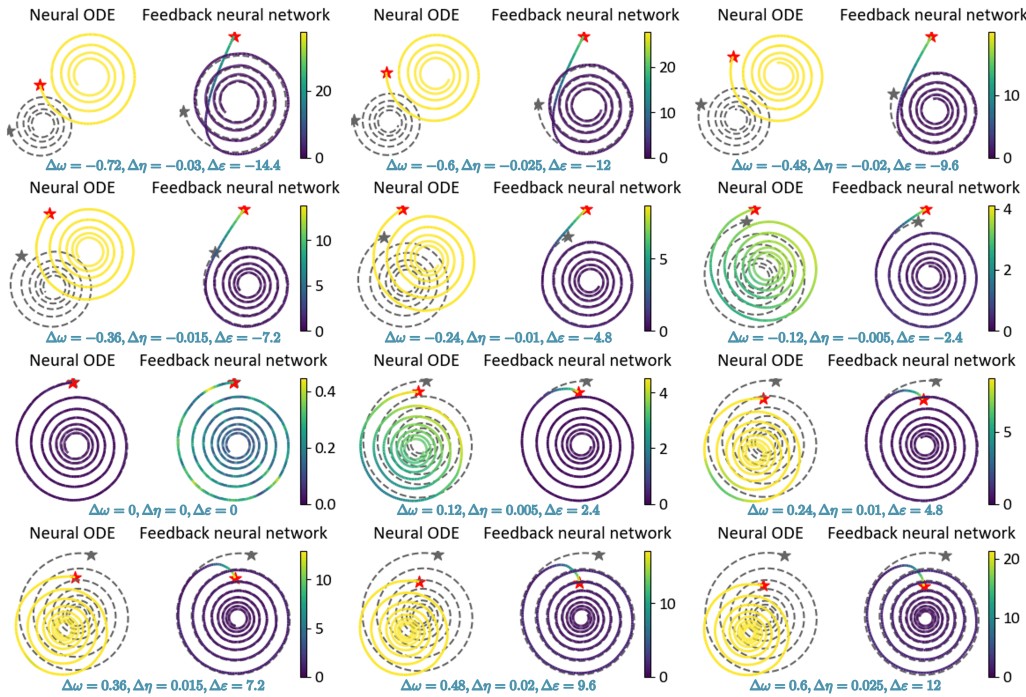

Figure S9: Test the linear feedback form on 12 randomized cases in the spiral curve example. It can be found that the developed feedback neural network shows better generalization performance as enabling the linear feedback unit.

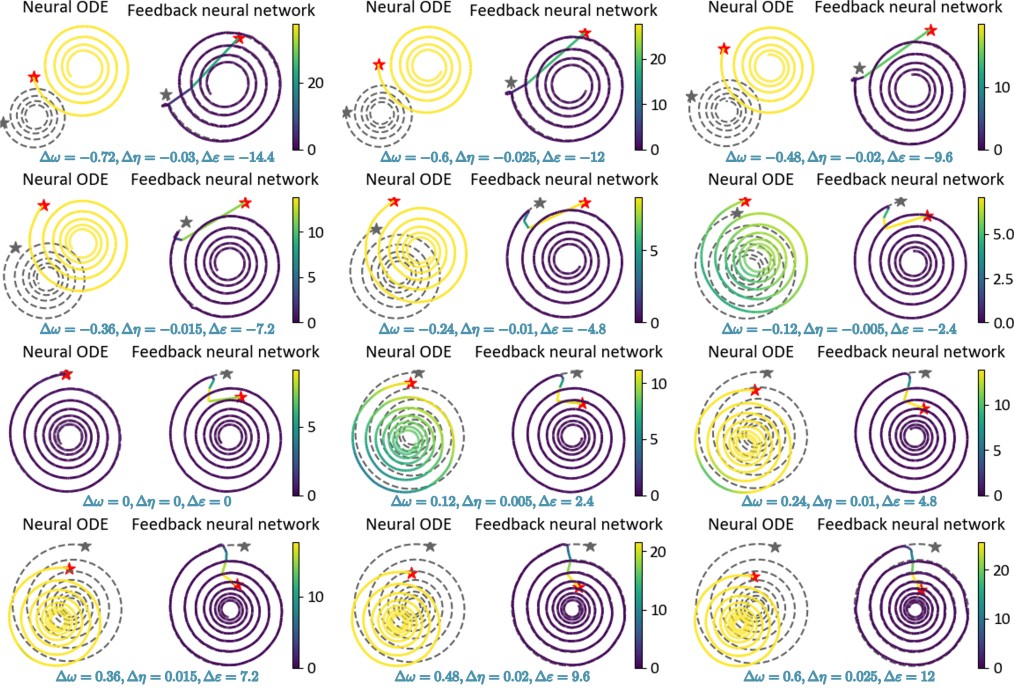

Figure S10: Test the neural feedback form on 12 randomized cases in the spiral curve example. The developed feedback neural network with the neural feedback unit shows better generalization performance than the neural ODE.

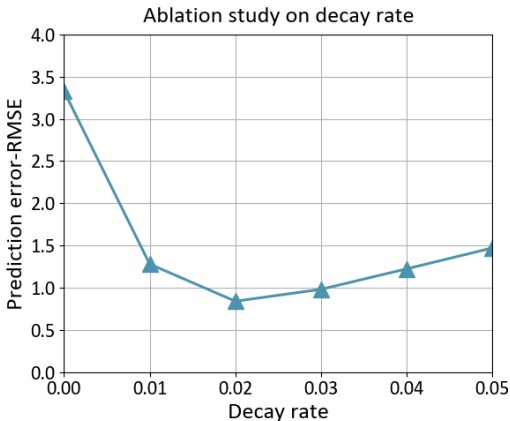

Figure S11: Ablation study on decay rate.

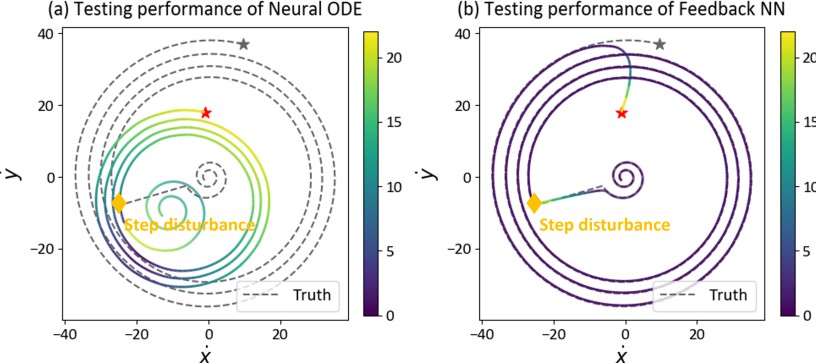

Figure S12: Test on step disturbance in the spiral curve example. The test setup is the same as Figure 5 except for the induce of step disturbance. As $t = 7\ s$ (denoted by a yellow diamond symbol), the latent dynamics is changed suddenly ($\omega : 3 \rightarrow 1,\ \eta : -0.05 \rightarrow -0.12,\ \varepsilon : 10 \rightarrow 5$). It can be seen that the proposed feedback neural network can attenuate the step disturbance quickly.

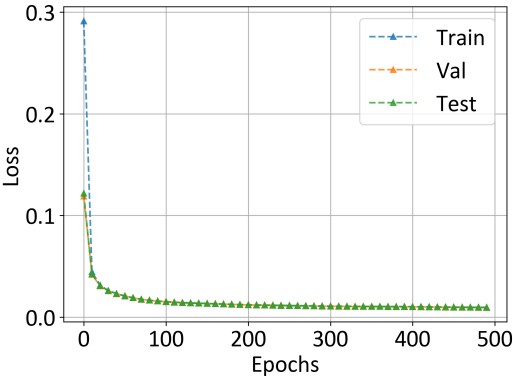

Figure S13: The training loss on the training set, the validation set, and the test set of the compared MLP augmented model.

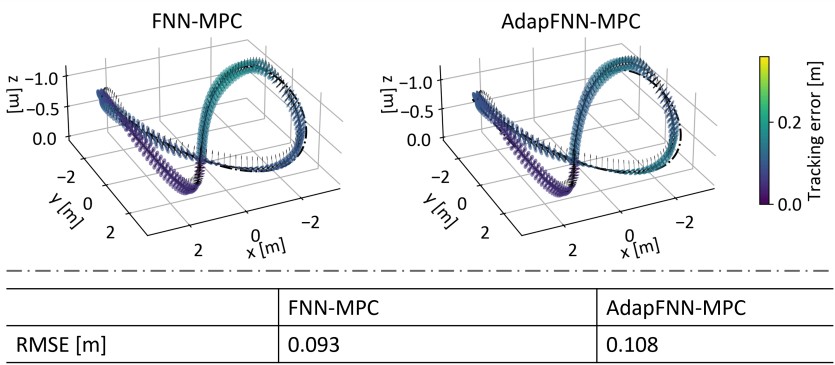

| | FNN-MPC | AdapFNN-MPC |
|---|---|---|
| RMSE [m] | 0.093 | 0.108 |

Figure S14: Tracking the *Lissajous* trajectory using MPC with adaptation-enhanced feedback neural network.

