# OpenReview forum: "Feedback Favors the Generalization of Neural ODEs"
_ICLR.cc/2025/Conference — ICLR 2025 Oral_

### Official Review · Reviewer_Lpz2 · 2024-11-02

**Soundness:** 2
**Presentation:** 4
**Contribution:** 3
**Rating:** 8
**Confidence:** 3

**Summary:**

This paper successfully enhanced the generalization capability of Neural ODEs by introducing feedback mechanisms, demonstrating significant performance improvements across multiple practical applications. Specifically, the authors proposed a Two-Degree-of-Freedom (Two-DOF) neural network architecture that combines linear and nonlinear feedback mechanisms, effectively correcting prediction errors and exhibiting stronger adaptability and generalization capability when dealing with complex and nonlinear dynamical systems. Overall, it provides a new perspective and solution for the generalization problem of neural networks in continuous-time prediction tasks.

**Strengths:**

- By combining linear and nonlinear (neural feedback) forms of feedback, an innovative network architecture is proposed that enhances generalization capability in unseen scenarios while maintaining accuracy on previous tasks. This design demonstrates innovation and practical value in current research. The experimental section covers trajectory prediction and quadrotor control, validating the method's superior performance.

- The paper provides theoretical analysis, proving system stability and convergence under linear feedback conditions. This establishes a solid theoretical foundation for the method's reliability and robustness. Additionally, the paper is well-structured with rigorous logic and clear writing. The figures and tables are intuitive. Detailed algorithm descriptions enhance the method's reproducibility.

- Through domain randomization techniques, the nonlinear neural feedback component is learned independently, demonstrating an effective way to improve generalization without significantly increasing training burden. This method effectively extends the applicability of feedback mechanisms when facing unknown uncertainties.

- The feedback loops can be easily integrated into existing trained Neural ODEs without requiring retraining, reducing implementation difficulties in practical applications and enhancing the method's feasibility and scalability. Furthermore, the method shows potential for online adjustment, further increasing its practical value.

- Generalization ability is a key challenge for neural networks in practical applications, and the paper effectively improves Neural ODE's generalization performance through feedback mechanisms, addressing the performance degradation issue of traditional methods in unseen tasks. This research has significant implications for neural network architecture design and practical engineering applications, showing broad application potential.

**Weaknesses:**

1. **Oversimplified Convergence Analysis (Section 3.2)**:
    - The convergence analysis is solely based on bounded learning residual error (Assumption 1), assuming an upper bound γ. However, in practical applications, system dynamics can be highly complex, making residual errors difficult to strictly bound and potentially time-varying or having more complex characteristics (such as unbounded growth or sudden disturbances). This idealized assumption limits the applicability of theoretical analysis in real scenarios.

2. **Manual Tuning of Feedback Gain Matrix \(L\)**:
    - The selection of feedback gain matrix \(L\) relies mainly on manual tuning, based on theoretical analysis and experimental results. This approach is both time-consuming and difficult to precise in complex systems, especially in high-dimensional systems or real-time changing environments where manual selection may not meet dynamic adjustment needs.

3. **Training Strategy Issues in Neural Network Feedback Section**:
    - The authors adopt a two-stage training method: first training the Neural ODE, then fixing parameters to train the feedback part. This approach may not be optimal as parameters of both networks might be coupled, preventing the overall model from achieving optimal performance under joint optimization. Could different training approaches be discussed?

4. **Insufficient Ablation Studies**:
    - Although the paper includes some ablation studies, they lack comprehensiveness and depth. For instance, there's insufficient in-depth analysis of individual components' impacts (like neural feedback network, linear and nonlinear feedback), and detailed sensitivity analysis of feedback gain matrix \(L\) and domain randomization parameters. Additionally, the limited variety of baseline models fails to cover more different types or improved Neural ODE variants, affecting the comprehensive demonstration of the method's advantages.

5. **Lack of Detailed Discussion on Computational Cost and Training Time**:
    - The paper's discussion of computational cost and training time is insufficient, failing to comprehensively address the proposed method's performance in terms of computational resources and training time compared to traditional methods.

**Questions:**

1. **Automated Design of Feedback Gain and Decay Rate**:
    - Could an adaptive mechanism be developed to dynamically adjust the feedback gain matrix \(L\) and decay rate \(\beta\) based on real-time prediction errors or system states? This would reduce the need for manual parameter tuning and enhance the method's flexibility and adaptability. Alternatively, provide more detailed documentation on the specific effects of feedback gain matrix \(L\) and decay rate \(\beta\).

2. **Comprehensive Ablation Studies and Sensitivity Analysis**:
    - Suggest adding ablation experiments on the independent effects of each component (such as neural feedback network, linear and nonlinear feedback), along with detailed sensitivity analysis of the feedback gain matrix \(L\) and domain randomization parameters. This would help better understand each part's contribution to overall performance.

3. **Compare with More Baseline Models**:
    - To more comprehensively demonstrate the advantages of the proposed method, it is recommended to compare with a wider range of baseline models, including other improved Neural ODE variants or feedback-based models, to validate its effectiveness across diverse scenarios.

4. **Detailed Description of Computational Cost and Training Time**:
    - Suggest adding a dedicated section in the paper to thoroughly discuss the performance of the proposed method in terms of computational resources and training time, including comparisons with traditional methods. This would help readers better evaluate its practical feasibility and efficiency.

---

> ### Author Response · Authors · 2024-11-19
> **Response to Reviewer Lpz2 (1/5)**
>
> We would like to express our greatest gratitude to Reviewer Lpz2 for the excellent comments. These comments are extremely valuable for improving both the technical quality and presentation of the manuscript. Based on your constructive advice, we have made our best efforts to thoroughly revise this paper. We sincerely hope that the revisions could satisfactorily meet your expectations. In addition, we have submitted a revised manuscript with an appendix where we mark all the suggested figures and analytics in magenta color.
>
> **Q1: Oversimplified Convergence Analysis (Section 3.2): The convergence analysis is solely based on bounded learning residual error (Assumption 1), assuming an upper bound γ. However, in practical applications, system dynamics can be highly complex, making residual errors difficult to strictly bound and potentially time-varying or having more complex characteristics (such as unbounded growth or sudden disturbances). This idealized assumption limits the applicability of theoretical analysis in real scenarios.**
>
> **Answer:** Yes, Assumption 1 indeed limits the performance of presented methods in the face of unbounded learning residuals.  In most uncertainty estimation works like [1-2], the derivative bounded assumption on unmodeled dynamics or learning residual error is usually held. In contrast, our work has gone one step further, and only the norm-bounded assumption on learning residual error is made, which is more acceptable and can handle step-like disturbance. In this revision, we further tested the performance of the proposed method in the face of sudden (step-like) dynamical change, as shown in Figure S12. It can be found that the feedback mechanism can attenuate the sudden disturbance quickly.
>
> The norm-bounded assumption on disturbance is also made in traditional control methods, like robust sliding mode control (SMC) [3]. However, the efficiency of SMC heavily relies on the sign function, which further depends on the extremely fast response speed and unlimited amplitude of the actuator. This requirement is difficult to meet in practical systems. Our method does not have this problem.
>
> As for unbounded learning residuals, we think it is still a major challenge in learning fields. It reveals that neural networks have completely lost the representational ability to target uncertainties. The best strategy may be retraining the networks based on fresh datasets, like an online continual learning mission [4]. The above discussion has been provided in Appendix A.2.1. Future work will keep this limitation in mind and try to follow the route of online continual learning to address it.
>
> **Revisions:** Figure S12 - test on sudden disturbances; discussion of assumptions in Appendix A.2.1.

---

> ### Author Response · Authors · 2024-11-19
> **Response to Reviewer Lpz2 (2/5)**
>
> **Q2:  Automated Design of Feedback Gain and Decay Rate: Could an adaptive mechanism be developed to dynamically adjust the feedback gain matrix ($L$) and decay rate ($\beta$) based on real-time prediction errors or system states? This would reduce the need for manual parameter tuning and enhance the method's flexibility and adaptability. Alternatively, provide more detailed documentation on the specific effects of feedback gain matrix ($L$) and decay rate ($\beta$).**
>
> **Answer:** Yes, the automated design of linear feedback gain is intriguing. That's also where we started trying to learn this feedback with neural structure in Section 4. Compared with the linear feedback, no prior gain tunning is required for the neural feedback at the cost of training cost.
>
> Here, we provide another raw idea to find optimal linear feedback gains without the definition of the decay rate. First, define the feedback gain of the $i$-th layer of multi-step prediction (Figure 3) as $L_i$. The gain may be different for each layer.  A bi-level optimization framework can be constructed to train neural ODE, while searching the optimal $L_i$
> $$
> \begin{array}{l}
> {\theta ^ * } = \mathop {\arg \min }\limits_\theta\  {J^{out}}(\theta ,{{L_{1,\cdots N,}} ^ * }): = \mathop {\arg \min }\limits_\theta  \sum\limits_{i \in {D_{nom}}} {l_1(x_i^ *  - {x_i})}
> \end{array}
> $$
> $$
> \begin{array}{l}
> s.t.\ \ \ {{L_{1,\cdots, N}}^ * } = \mathop {\arg \min }\limits_{{L_{1,\cdots, N}}}\  {J^{in}}(\theta ,{L_{1,\cdots N,}} ): = \mathop {\arg \min }\limits_{L_{1,\cdots, N}}  \sum\limits_{j \in {D_{dom}}} {l_2(x_j^ *  - {x_j})}
> \end{array}
> $$
> where feedforward networks are parameterized by $\theta$,  $L_{1,\cdots,N}$ denotes all linear observer gains, ${J^{out}}$ and ${J^{in}}$ refer to outer and inner objectives respectively, $l_1$ and $l_2$ are defined to quantify the state differences between model rollout $x_i$ and real-world state $x_i^ *$, ${D_{nom}}$ represents training trajectories collected from the nominal case, and ${D_{dom}}$ represents training trajectories collected from the disturbed cases through domain randomization. $L_i$ could be a log-Cholesky parameterization to make sure its positive definite feature during learning. Note that the inner objective is differentiable with respect to $L_{1,\cdots N,}$. In future work, we will refine the above bi-level optimization algorithm. We have stated this point in Limitation.
>
> Moreover, combined with your comment Q4, we have conducted ablation studies on the feedback gain matrix and decay rate. In practice, the tunning of feedback gain and decay rate is very intuitive. They can be increased slowly from a small value until the critical value with the best estimation performance is reached.
>
> **Revisions:** Related description of above bi-level optimization in Limitation; ablation studies on linear feedback gain (Section 3.4), and decay rate (Appendix 6.3, Figure S11).
>
> **Q3: Training Strategy Issues in Neural Network Feedback Section: The authors adopt a two-stage training method: first training the Neural ODE, then fixing parameters to train the feedback part. This approach may not be optimal as parameters of both networks might be coupled, preventing the overall model from achieving optimal performance under joint optimization. Could different training approaches be discussed?**
>
> **Answer:**  We are glad to discuss the two-stage training strategy with you. One important motivation for this work is that we want to separate the accuracy and generalization features of neural networks. Within the feedback neural network, the accuracy feature is given to the feedforward part, while the generalization feature is specialized to the feedback part.
>
> From another point of view, traditional neural network *vs* proposed feedback neural network is similar to robust control *vs* disturbance observer-based control in the control field. The robust control is a single-DOF structure [5], where contradictions exist among different performance requirements (e.g., tracking *vs* disturbance rejection). The disturbance observer-based control is a 2-DOF framework that can equip both tracking and disturbance rejection performance [6]. The separation of controller and observer parameters (i.e., the principle of separability) is practical. This idea inspires us to decouple the neural network.

---

> > ### Author Response · Authors · 2024-11-19
> > **Response to Reviewer Lpz2 (3/5)**
> >
> > Indeed, the current two-stage training method cannot reflect the coupled information. In future work, following the response to your last comment Q2, we will try to establish a bi-level optimization framework to jointly optimize these two networks. A preliminary formalization is provided here
> > $$
> > \begin{array}{l}
> > {\theta ^ * } = \mathop {\arg \min }\limits_\theta\  {J^{out}}(\theta ,{\xi ^ * }): = \mathop {\arg \min }\limits_\theta  \sum\limits_{i \in {D_{nom}}} {l_1(x_i^ *  - {x_i})}
> > \end{array}
> > $$
> > $$
> > \begin{array}{l}
> > s.t.\ \ \ {\xi ^ * } = \mathop {\arg \min }\limits_\xi\  {J^{in}}(\theta ,\xi ): = \mathop {\arg \min }\limits_\xi  \sum\limits_{j \in {D_{dom}}} {l_2(x_j^ *  - {x_j})}
> > \end{array}
> > $$
> > where feedforward and feedback networks are parameterized by $\theta$ and $\xi$ respectively, ${J^{out}}$ and ${J^{in}}$ refer to outer and inner objectives respectively, $l_1$ and $l_2$ are defined to quantify the state differences between model rollout $x_i$ and real-world state $x_i^ *$, ${D_{nom}}$ represents training trajectories collected from the nominal case, and ${D_{dom}}$ represents training trajectories collected from the disturbed cases through domain randomization. Moreover, the loss $l_2$ could be related to some noise-related objectives, where the amplifying effects of the continuously injected Gaussian noises can be approximately computed with local linearized dynamics, bringing us an insight into the robustness of the feedback mechanism. In this way, each network is trained to take into account the impact of the other. In future work, we will refine the above bi-level optimization algorithm.
> >
> > **Revision:** Related description in Limitation.
> >
> > **Q4: Comprehensive Ablation Studies and Sensitivity Analysis: Suggest adding ablation experiments on the independent effects of each component (such as neural feedback network, linear and nonlinear feedback), along with detailed sensitivity analysis of the feedback gain matrix ($L$) and domain randomization parameters. This would help better understand each part's contribution to overall performance.**
> >
> > **Answer:** In the revised manuscript, we have supplemented several ablation studies to show the effectiveness of each part.
> >
> > - Ablation study on linear feedback gain: In Section 3.4, we test prediction errors of the spiral curve with different levels of feedback gains and parameter uncertainties. Two phenomena can be observed from the heatmap. The one is that the prediction error increases with the level of uncertainty. The other is that the prediction error decreases with the gain at the beginning fitting theory analysis. However, due to noise amplification, the prediction error worsens if the gain is set too large.
> >
> > - Ablation study on linear feedback unit: We test the performance of correcting the latent dynamics of spiral curves at $12$ different levels of learning residuals, with or without linear feedback unit. The parameter uncertainties cover $\Delta \omega= \\{-0.72: +0.12: 0.6\\}$, $\Delta \eta= \\{-0.03: +0.005: 0.025\\}$, $\Delta {\varepsilon}= \\{-14.4: +2.4: 12\\}$. All compared results are summarized in Figure S9, which indicates the effectiveness of the linear feedback unit.
> > - Ablation study on neural feedback unit: Similar to the last test, we further test the performance of the neural feedback unit at $12$ different levels of learning residuals. The induced uncertainties are exactly the domain randomization parameters during learning. All compared results are summarized in Figure S10. It can be found that the developed feedback neural network shows better generalization performance by enabling the neural feedback unit. Moreover, the neural/nonlinear feedback part cannot be trained without domain randomization. They are bound together.
> > - Ablation study on decay rate: The performance of the decay rate is examined in the spiral curve example. In tests, the decay rate is set as  $\beta= \\{0: +0.01: 0.06\\}$ in sequence, and the multi-step prediction errors (Figure 5(g)) are calculated in RMSE. The test results are shown in the Figure S11. It can be seen that the prediction error decreases with the increase of $\beta$ at the beginning due to noise mitigation. However, as $\beta$ continues to increase, the convergence time becomes slower, leading to a gradual increase in prediction error.
> >
> > **Revisions:** Ablation studies on linear feedback gain (Section 3.4), linear feedback unit (Appendix 6.1, Figure S9), neural/nonlinear feedback unit (Appendix 6.2, Figure S10), and decay rate (Appendix 6.3, Figure S11).

---

> ### Author Response · Authors · 2024-11-19
> **Response to Reviewer Lpz2 (4/5)**
>
> **Q5: Compare with More Baseline Models: To more comprehensively demonstrate the advantages of the proposed method, it is recommended to compare with a wider range of baseline models, including other improved Neural ODE variants or feedback-based models, to validate its effectiveness across diverse scenarios.**
>
> **Answer:** Following your advice, we have compared two previous learning-based methods for the quadrotor example. One is the feedforward neural network-based residual model [7] which employs the fully connected neural network to learn aerodynamic drag. The other is the adaptive neural network-based residual model [8] in which the last layer of the neural network is regarded as a weighted vector, being adjusted adaptively according to real-time state feedback.  The RMSE of all implemented methods in the quadrotor example are summarized in the following table, where AdapNN-MPC refers to the adaptive neural network method [8] updating the last layer online, and MLP-MPC represents the feedforward neural network-based method [7].
>
> |          | Nomi-MPC | Neural-MPC | FB-MPC | MLP-MPC | AdapNN-MPC | FNN-MPC   |
> | -------- | -------- | ---------- | ------ | ------- | ---------- | --------- |
> | RMSE [m] | 0.248    | 0.167      | 0.203  | 0.182   | 0.151      | **0.093** |
>
> Figure 9 shows the trajectory tracking performance. It can be seen that the performance of MLP-MPC is relatively unsatisfactory compared with that of Neural-MPC. The reason can be attributed to its single-step training manner instead of the multi-step one of the Neural-MPC, leading to a poor multi-step prediction required by MPC. Due to the adaptive ability of the last layer, AdapNN-MPC can handle a certain level of uncertainty. Finally, FNN-MPC achieves the best tracking performance. The reason can be attributed to the multi-step prediction algorithm of the feedback neural network, which improves the prediction accuracy subject to multiple uncertainties.
>
> **Revisions:** Figure 9 - test results of the quadrotor example; descriptions of compared methods and results in Section 5.2.2 and Appendix A.5.4.
>
> **Q6: Detailed Description of Computational Cost and Training Time: Suggest adding a dedicated section in the paper to thoroughly discuss the performance of the proposed method in terms of computational resources and training time, including comparisons with traditional methods. This would help readers better evaluate its practical feasibility and efficiency.**
>
> **Answer:** Thanks for this valuable comment.  In this revision, we have added a section (Appendix A.7) to illustrate the computational costs of the presented methods.
>
> For the training of neural ODEs, two strategies are employed in this work: the adjoint sensitive method developed in [9] without considering external inputs, and our own alternative training method developed in Appendix A.1 concerning external inputs. The adjoint sensitive method is utilized in the spiral curve and irregular object examples, and its computational resource and training time are the same as [9]. The alternative training method concerning external inputs is employed in the quadrotor example to learn residual dynamics with external inputs. It takes around 30 mins to run 50 epochs on a laptop with 13th Gen Intel(R) Core(TM) i9-13900H. The alternative training method is derived from the view of optimal control, and its computational resource and training time are comparable to the adjoint sensitive method theoretically.
>
> Two feedback forms are presented. No prior training is required for the linear feedback form, showing its advantage over traditional learning-based generalization methods. Moreover, the linear feedback consists of several analytic equations, consuming almost no computing resources. As for the neural feedback form, due to the optimization problem being non-convex, a satisfactory result usually takes 10 mins to 1 hour of training time on a laptop with Intel(R) Core(TM) Ultra 9 185H 2.30 GHz.
>
> **Revision:** Appendix A.7 - Training cost.

---

> ### Author Response · Authors · 2024-11-19
> **Response to Reviewer Lpz2 (5/5)**
>
> **Reference**
>
> [1] O’Connell, Michael, et al. "Neural-fly enables rapid learning for agile flight in strong winds." *Science Robotics* 7.66 (2022): eabm6597.
>
> [2] Jia, Jindou, et al. "EVOLVER: Online Learning and Prediction of Disturbances for Robot Control." *IEEE Transactions on Robotics* (2023).
>
> [3] Yang, Jong-Min, and Jong-Hwan Kim. "Sliding mode control for trajectory tracking of nonholonomic wheeled mobile robots." *IEEE Transactions on Robotics and Automation* 15.3 (1999): 578-587.
>
> [4] Y. Ghunaim, et al. "Real-time evaluation in online continual learning: A new hope." *Proceedings of the IEEE/CVF Conference on Computer Cision and Pattern Recognition*. 2023.
>
> [5] Chen, Wen-Hua, et al. "Disturbance-observer-based control and related methods—An overview." *IEEE Transactions on Industrial Electronics* 63.2 (2015): 1083-1095.
>
> [6] Li, Xian, et al. "Feedforward control with disturbance prediction for linear discrete-time systems." *IEEE Transactions on Control Systems Technology* 27.6 (2018): 2340-2350.
>
> [7]  Alessandro Saviolo and Giuseppe Loianno. "Learning quadrotor dynamics for precise, safe, and agile flight control." *Annual Reviews in Control* 55 (2023): 45-60.
>
> [8] Y. Cheng, et al. "Human motion prediction using semi-adaptable neural networks." *American Control Conference (ACC).* 2019.
>
> [9] Chen, Ricky TQ, et al. "Neural ordinary differential equations." *Advances in Neural Information Processing Systems*. 2018.

---

> ### Author Response · Authors · 2024-11-21
> **Looking forward to your reply**
>
> Dear Reviewer Lpz2,
>
> We have provided new experiments and discussions according to your valuable suggestions, which have been absorbed into the revised manuscript. We hope that the new manuscript is made to be stronger with your suggestions.
>
> As the rebuttal deadline is approaching, we sincerely look forward to your reply. Feel free to let us know if you have any other concerns. Thanks so much for your time and effort!
>
> Best Regards,
>
> Authors of Submission 8713

---

> > ### Comment · Reviewer_Lpz2 · 2024-11-22
> >
> > Thank you for the contributions. I checked your work and updated the score.

---

> > > ### Author Response · Authors · 2024-11-22
> > > **Appreciation for the constructive comments**
> > >
> > > We sincerely appreciate your constructive comments which help us improve our manuscript. Thanks for your time and effort!

---

### Official Review · Reviewer_a2wg · 2024-11-03

**Soundness:** 3
**Presentation:** 3
**Contribution:** 4
**Rating:** 8
**Confidence:** 4

**Summary:**

This paper introduces closed-loop feedback neural ODEs which offer better generalization performance compared to open-loop neural ODEs. A linear state-error feedback term is introduced to compensate for prediction inaccuracies in the closed-loop error mechanism. This feedback term is shown to improve the generalization performance which was poor in the previous open-loop neural ODE formulations.

**Strengths:**

I believe this paper makes a strong contribution. The paper is well-written and easy to follow. The technical development with the use of closed-loop state error feedback term is well-motivated. I found the paper an interesting read.

**Weaknesses:**

Despite the significant contributions, there are quite a few mathematical errors and a general lack of mathematical rigor in the paper. These errors might be minor and easy to resolve, I suggest the authors address the following issues.

1. Equation (7) appears erroneous. Combining (4) and (6) should actually give the term $L(\bar{x}(t)-\hat{x}(t))$, not $L(x(t)-\hat{x}(t))$. This error gets carried forward to Eq. (9) of the convergence analysis, thus potentially invalidating Theorem 1. As far as I can help, this issue can be avoided redefining Eq. (6) in terms of $x(t)$ instead of $\bar{x}(t)$. In that case, Eq. (8) needs to be adjusted accordingly. Alternatively, the authors can add and subtract $Lx(t)$ term in the derivation of Eq. (9), which should result in an extra $L(x-\bar{x})$ term in (9). This term will need to be bounded and that bound would need to be considered in the ensuing Lyapunov-based stability analysis to obtain the correct error bounds. Either way, the mathematical analysis needs revisions to address the error in Eq. (7).

2. The development in Eq. (4)-(8) is in a discrete-time setting but the convergence analysis in Section 3.2 is in a continuous-time setting. To be consistent, the authors are suggested to either reformulate the development in Section 3.1 using continuous-time integrals instead of discrete-time summations, or rederive the convergence analysis in Section 3.2 using discrete-time Lyapunov theorem or contraction analysis. I believe this should be easy to address.

3. Although the proof of Theorem 1 is correct, it appears the analysis is not performed in the most efficient manner. Specifically, the use of Young's inequality in Eq. (2) is not the most efficient because it results in the $\lambda_m(L)-\frac{1}{2}$ term which carries forward through the analysis. The problem with this term is that the gain condition $\lambda_m(L)>frac{1}{2}$ would need to be imposed to enure an exponential decay. Such a gain condition is missing in the paper. However, changing the analysis in Eq. (20) would remove the need for a gain condition. I suggest the authors use the following form of Young's inequality instead

$$ \tilde{x}^T \Delta f(t) = \sqrt{\lambda_m(L)}||\tilde{x}||\frac{\gamma}{\sqrt{\lambda_m(L)}}=\frac{\lambda_m(L)||\tilde{x}||^2}{2}+\frac{\gamma^2}{2\lambda_m(L)}.$$

This would result in

$$\dot{V} \leq -\frac{\lambda_m(L)||\tilde{x}||^2}{2}+\frac{\gamma^2}{2\lambda_m(L)}.$$

This formulation would eliminate the need of an extra gain condition, while also obtaining a tighter bound in the $\delta$ term.

4. The purely time-dependent exponentially decaying gain schedule for the feedback gain in Eq. (11) can be problematic in terms of the performance, because the theoretical error bounds in Theorem 1 contain $\lambda_m(L)$ in the denominator. The bounds get enormous with the exponential decay, thus essentially making the neural ODE open-loop with time. There are better ways to schedule the gain using closed-loop feedback. The authors are recommended to look into recursive least squares algorithms that are popular in the adaptive control or Kalman filtering literature, where the gain decreases as the estimator gains more information instead of scheduling it to always decay with time.

5. This is minor but the technical writing throughout the paper lacks mathematical rigor. Specifically, the mathematical entities (states, functions, constants etc.) are not formally defined. For example, the domain of x can be precisely defined, i.e., $x(t)\in\mathbb{R}^n$. The function $f$ can be defined formally as $f:\mathbb{R}^n\times\mathbb{R}^m\times\mathbb{R}\to \mathbb{R}^n$. The notations $f_\{neural}(t)$ and $\hat{f}_\{neural}(t)$ (which are not formally defined) are confusing because they take only time as the argument. However, they also seem to depend on x and \hat{x}. The term $L$is defined to be a constant, but it should actually be a positive-definite matrix. The concatenated state $z$ needs to be defined similarly with an appropriate domain. In the statement of Theorem 1, instead of stating "the state observation error and its derivative can converge to bounded sets exponentially, which upper bounds can be regulated by the feedback gain L", the authors should explicitly state the obtained exponentially convergent bounds.

6. Minor issue, but Figure 1 seems to indicate the error is being passed through another neural network instead of a linear feedback gain, which is not the case in this paper. The figure needs to be revised replacing this second neural network with a -L gain block.

EDIT: Increasing the score to 8 after author responses.

**Questions:**

I do not have further questions or suggestions beyond the comments I made above. The authors are suggested to carefully examine all of my above comments. I commend the authors for this interesting work and look forward to reading their responses.

---

> ### Author Response · Authors · 2024-11-19
> **Response to Reviewer a2wg (1/3)**
>
> Thanks to Reviewer a2wg for the recognition of our work. We would like to express our sincerest gratitude for the time and efforts spent on handling and reviewing our manuscript, especially the mathematical derivation. All your comments have been taken into account thoroughly during the revision process. In addition, we have submitted a revised manuscript with an appendix where we mark all the suggested figures and analytics in magenta color.
>
> **Q1: Equation (7) appears erroneous. Combining (4) and (6) should actually give the term $L(\bar{x}(t)-\hat{x}(t))$, not $L({x}(t)-\hat{x}(t))$. This error gets carried forward to Eq. (9) of the convergence analysis, thus potentially invalidating Theorem 1. As far as I can help, this issue can be avoided by redefining Eq. (6) in terms of ${x}(t)$ instead of $\bar{x}(t)$. In that case, Eq. (8) needs to be adjusted accordingly. Alternatively, the authors can add and subtract $L{x}(t)$ term in the derivation of Eq. (9), which should result in an extra$L({x}(t)-\bar{x}(t))$term in (9). This term will need to be bounded and that bound would need to be considered in the ensuing Lyapunov-based stability analysis to obtain the correct error bounds. Either way, the mathematical analysis needs revisions to address the error in Eq. (7).**
>
> **Answer:** The derivation procedure of Eq. (7) has been checked carefully and we think the previous Eq. (7) is right. For your convenience, the derivation procedure is detailed here. From Eq. (4), we have
> $${\hat f_{neural}}(t) = {f_{neural}}(t) +  \sum\limits_{i = 0}^{k-1} {{L}\left( {{x}\left( {{t_i}} \right) - {\bar{x}}\left( {{t_i}} \right)} \right)} + {L}({x}\left( {{t_k}} \right)-{\bar{x}}\left( {{t_k}} \right)).$$
> The part $\sum\limits_{i = 0}^{k-1} {{L}\left( {{x}\left( {{t_i}} \right) - {\bar{x}}\left( {{t_i}} \right)} \right)}$ in above equation can be obtained from Eq. (6), in the form $\sum\limits_{i = 0}^{k-1} {{L}\left( {{x}\left( {{t_i}} \right) - {\bar{x}}\left( {{t_i}} \right)} \right)} = {L}({\bar{x}}\left( {{t}} \right)-{\hat{x}}\left( {{t}} \right))$. Thus, it can be further obtained that
> $$
> {\hat f_{neural}}(t) = {f_{neural}}(t) +{L}({\bar{x}}\left( {{t}} \right)-{\hat{x}}\left( {{t}} \right))+ {L}({x}\left( {{t_k}} \right)-{\bar{x}}\left( {{t_k}} \right))
> $$
> $$
> {\hat f_{neural}}(t) =  {f}_{neural}(t) -L{\hat{x}}\left( {{t}} \right) + {L}{x}\left( {{t_k}} \right)
> $$
>
> $$
> {\hat f_{neural}}(t) =  {f}_{neural}(t) + {L}({x}\left( {{t_k}} \right)-{\hat{x}}\left( {{t}} \right))
> $$
> which is exactly Eq. (7).
> In the revised paper, we also checked other mathematical derivations, and several minor issues (lines 118 and 167) have been corrected. Thanks for the careful check.
>
> **Revisions:** Lines 118 and 167.
>
> **Q2: The development in Eq. (4)-(8) is in a discrete-time setting but the convergence analysis in Section 3.2 is in a continuous-time setting. To be consistent, the authors are suggested to either reformulate the development in Section 3.1 using continuous-time integrals instead of discrete-time summations or rederive the convergence analysis in Section 3.2 using discrete-time Lyapunov theorem or contraction analysis. I believe this should be easy to address.**
>
> **Answer:** Thanks for this valuable advice. In this revision, we have provided the convergence analysis in Appendix A.2.2 using the discrete-time theorem.  The reason for the discrete-time form of Eqs. (4)-(8) is that we intended to introduce the feedback idea in an intuitive and achievable way. Thus, we decide to maintain the discrete-time form of Eqs. (4)-(8) in the revisied paper.
>
> **Revision:** Appendix A.2.2 - Discrete-time stability.

---

> ### Author Response · Authors · 2024-11-19
> **Response to Reviewer a2wg (2/3)**
>
> **Q3: Although the proof of Theorem 1 is correct, it appears the analysis is not performed in the most efficient manner. Specifically, the use of Young's inequality in Eq. (2) is not the most efficient because it results in the $\lambda_m({L})>\frac{1}{2}$ term which carries forward through the analysis. The problem with this term is that the gain condition $\lambda_m({L})>\frac{1}{2}$ would need to be imposed to ensure an exponential decay. Such a gain condition is missing in the paper. However, changing the analysis in Eq. (20) would remove the need for a gain condition. I suggest the authors use the following form of Young's inequality instead**
> $$
> \tilde{x}^{T} \Delta f(t)\le\sqrt{\lambda_{m}(L)}\|\tilde{x}\| \frac{\gamma}{\sqrt{\lambda_{m}(L)}}\le\frac{\lambda_{m}(L)\|\tilde{x}\|^{2}}{2}+\frac{\gamma^{2}}{2 \lambda_{m}(L)} .
> $$
> **This would result in**
> $$
> \dot{V} \leq-\frac{\lambda_{m}(L)\|\tilde{x}\|^{2}}{2}+\frac{\gamma^{2}}{2 \lambda_{m}(L)} .
> $$
> **This formulation would eliminate the need of an extra gain condition, while also obtaining a tighter bound in the $\delta$ term.**
>
> **Answer:** It is a constructive comment. In this revision, we have used the suggested Young's inequality and yielded a tighter convergence bound. Thanks for this advice.
>
> **Revisions:** Appendix A.2 - Proof of Theorem 1; a related statement about the gain condition in section 3.4.
>
> **Q4: The purely time-dependent exponentially decaying gain schedule for the feedback gain in Eq. (11) can be problematic in terms of the performance, because the theoretical error bounds in Theorem 1 contain$\lambda_m({L})$ in the denominator. The bounds get enormous with the exponential decay, thus essentially making the neural ODE open-loop with time. There are better ways to schedule the gain using closed-loop feedback. The authors are recommended to look into recursive least squares algorithms that are popular in the adaptive control or Kalman filtering literature, where the gain decreases as the estimator gains more information instead of scheduling it to always decay with time.**
>
> **Answer:** Yes, we also noticed this theoretical drawback. In this revision, we have supplemented several ablation studies (Appendix A.6) including the decay rate (Figure S11).  Not surprisingly, the prediction error decreases with the increase of $\beta$ at the beginning due to noise mitigation. As $\beta$ continues to increase, the convergence time becomes slower, leading to a gradual increase in prediction error.
>
> Here, we provide another raw idea to find optimal linear feedback gains without the definition of the decay rate. First, define the feedback gain of the $i$-th layer of multi-step prediction (Figure 3) as $L_i$. The gain may be different for each layer.  A bi-level optimization framework can be constructed to train neural ODE, while searching the optimal $L_i$
> $$
> \begin{array}{l}
> {\theta ^ * } = \mathop {\arg \min }\limits_\theta\  {J^{out}}(\theta ,{{L_{1,\cdots N,}} ^ * }): = \mathop {\arg \min }\limits_\theta  \sum\limits_{i \in {D_{nom}}} {l_1(x_i^ *  - {x_i})}
> \end{array}
> $$
> $$
> \begin{array}{l}
> s.t.\ \ \ {{L_{1,\cdots, N}}^ * } = \mathop {\arg \min }\limits_{{L_{1,\cdots, N}}}\  {J^{in}}(\theta ,{L_{1,\cdots N,}} ): = \mathop {\arg \min }\limits_{L_{1,\cdots, N}}  \sum\limits_{j \in {D_{dom}}} {l_2(x_j^ *  - {x_j})}
> \end{array}
> $$
> where feedforward networks are parameterized by $\theta$,  $L_{1,\cdots,N}$ denotes all linear observer gains, ${J^{out}}$ and ${J^{in}}$ refer to outer and inner objectives respectively, $l_1$ and $l_2$ are defined to quantify the state differences between model rollout $x_i$ and real-world state $x_i^ *$, ${D_{nom}}$ represents training trajectories collected from the nominal case, and ${D_{dom}}$ represents training trajectories collected from disturbed cases through domain randomization. $L_i$ could be a log-Cholesky parameterization to make sure its positive definite feature during learning. Note that the inner objective is differentiable with respect to $L_{1,\cdots N,}$. In future work, we will refine the above bi-level optimization algorithm. We have stated this point in Limitation.
>
> I would also like to thank the reviewer for the idea about the recursive least squares algorithm. We will try to use it where appropriate.
>
> **Revision:** Related description in Limitation.

---

> ### Author Response · Authors · 2024-11-19
> **Response to Reviewer a2wg (3/3)**
>
> **Q5: This is minor but the technical writing throughout the paper lacks mathematical rigor. Specifically, the mathematical entities (states, functions, constants etc.) are not formally defined. For example, the domain of $x$ can be precisely defined, i.e., $x(t) \in \mathbb{R}^n$. The function $f$ can be defined formally as $f:\mathbb{R}^n \times \mathbb{R}^m \times\ \mathbb{R} \rightarrow \mathbb{R}^n$ . The notations $f_{neural}(t)$ and$\hat{f}_{neural}(t)$ (which are not formally defined) are confusing because they take only time as the argument. However, they also seem to depend on $x$ and $\hat{x}$. The term $L$ is defined to be a constant, but it should actually be a positive-definite matrix. The concatenated state $z$ needs to be defined similarly with an appropriate domain. In the statement of Theorem 1, instead of stating "the state observation error and its derivative can converge to bounded sets exponentially, which upper bounds can be regulated by the feedback gain $L$", the authors should explicitly state the obtained exponentially convergent bounds.**
>
> **Answer:** By following your advice, all notations used in this paper have been formally defined, including $x$, $I$, $f$, $\Delta f$, $ f_{neural}$,$\hat{f}_{neural}$, $L$, ${\bar{x}}$, $T_s$, $\hat{x}$, $\gamma$, $z$, $h$, and other symbols used in Appendix. The obtained convergent bound has also been clearly stated in Theorem 1. In addition, a test on the theoretical convergent bound has also been carried out for the spiral curve example, as shown in Figure S1. The result shows that the state observation error can converge to the theoretical bounded set. It can also be found that the theoretical bounded set is relatively conservative, as the result of the sufficiency of *Lyapunov* theorem.
>
> **Revisions:** All defined notations are marked in magenta; Theorem 1; Figure S1 - test of convergence set.
>
> **Q6: Minor issue, but Figure 1 seems to indicate the error is being passed through another neural network instead of a linear feedback gain, which is not the case in this paper. The figure needs to be revised by replacing this second neural network with a -$L$ gain block.**
>
> **Answer:** Two kinds of feedback forms are developed in this work, a linear form (Section 3) and a nonlinear neural form (Section 4). The previous Figure 1 mainly portrayed the nonlinear neural form. Following your advice, we have provided all proposed forms in Figure 1 in the revised manuscript.
>
> **Revision:** Figure 1.

---

> > ### Comment · Reviewer_a2wg · 2024-11-19
> > **One minor comment remaining**
> >
> > I thank the authors for their responses. Most of my comments have been addressed. I still have one minor comment which should be easy to address. The statement of Theorem 1 is not precise as it refers to $\gamma/\lambda_m (L)$ and the other term as sets.  These terms are not sets but radii of these sets. Please explicitly define appropriate sets, .e.g., $B= \\{ x \in \mathbb{R}^n : ||x|| \leq  \gamma/\lambda_m (L) \\}$ and refer to these sets explicitly in the Theorem statement to show convergence. I will update my score after this last comment is addressed.

---

> > > ### Author Response · Authors · 2024-11-20
> > > **Thank you for the response**
> > >
> > > Dear reviewer,
> > >
> > > Thank you for your response and further pointing out an expression issue. We have stated the suggested convergence sets formally in Theorem 1 and Appendix A.2.1. For your convenience, we rephrase Theorem 1 here.
> > >
> > > **Theorem 1.** Consider the nonlinear system (1). Under the linear state feedback (7) and the bounded Assumption 1, the state observation error ${{\tilde x}}(t)$ and its derivative ${{\dot{\tilde x}}}(t)$ (i.e., ${{\tilde f}}(t)$) can exponentially converge to bounded sets $\mathcal{B}_1 = \\{{{\tilde x}}(t)\in\mathbb{R}^n:||{{\tilde x}}(t)||\le{\gamma}/{{\lambda_m({L})}}\\}$ and $\mathcal{B}_2 = \\{{{\dot{\tilde x}}}(t)\in\mathbb{R}^n:||{{\dot{\tilde x}}}(t)||\le{\gamma}\lambda_M({L})/{{\lambda_m({L})}} + \gamma\\}$, respectively, which can be regulated by ${L}$.
> > >
> > > We sincerely appreciate your constructive comments and prompt response which help us improve our paper.

---

> ### Comment · Reviewer_a2wg · 2024-11-20
>
> Thanks for the response. I have updated the score to 8. Another minor issue I noticed: in Section 6.3, the word `adaption` should be correctly spelt as `adaptation`. Also now that the authors have added discussion on real-time updates based on other reviewer comments, the following work which performs real-time all-layer updates can also be incorporated into the discussion:
>
> Patil, O.S., Le, D.M., Greene, M.L. and Dixon, W.E., 2021. Lyapunov-derived control and adaptive update laws for inner and outer layer weights of a deep neural network. IEEE Control Systems Letters, 6, pp.1855-1860.
>
> Also, I am not sure if the statement "``the performance of this strategy is limited by the inherent shortcoming of adaptive control, i.e., the time-invariant assumption on the weighted vector``" is accurate. Adaptive control methods can track piecewise constant parameters and are known to be robust to slow parameter variations (see Slotine's Applied Nonlinear Control book for reference). Maybe a correct limitation the authors can point would be that real-time weight updates require extra computation; I would also let other reviewers weigh in their opinion about this statement. Regardless, it is not necessary to state any limitation of adaptive control methods in this section. I believe stating that adaptive neural ODEs with feedback can be investigated in future work would make a convincing argument.

---

> > ### Author Response · Authors · 2024-11-21
> > **Thank you for the response**
> >
> > Thanks for this valuable discussion on the neural network-based adaptive control.
> >
> > In this revision, 1) the 'adaption' problem has been addressed; 2) the previous important work [1] has been covered in Section 6.3; 3) the limitation about the time-invariant assumption on the weighted vector has been removed to avoid ambiguity.
> >
> > We also investigated some literature on adaptive control with time-varying parameters [2-4], which shows that a lot of work has been focused on this case, such as the robust adaptive control [2] and the congelation of variables method [4]. Most work can guarantee bounded convergence with a slow or bounded assumption on parameters. We are unsure if there is work that can guarantee zero-error convergence, although we also haven't achieved it. To ensure rigor, we decided to remove the related statement in the manuscript as you suggested. Moreover, the potential for adaptive neural ODEs to be combined with feedback mechanisms has also been described in Section 6.3.
> >
> > **Reference**
> >
> > [1] Patil, Omkar Sudhir, et al. "Lyapunov-derived control and adaptive update laws for inner and outer layer weights of a deep neural network." IEEE Control Systems Letters 6 (2021): 1855-1860.
> >
> > [2] Annaswamy, Anuradha M., and Alexander L. Fradkov. "A historical perspective of adaptive control and learning." Annual Reviews in Control 52 (2021): 18-41.
> >
> > [3] Guo, Kai, and Yongping Pan. "Composite adaptation and learning for robot control: A survey." Annual Reviews in Control 55 (2023): 279-290.
> >
> > [4] Chen, Kaiwen, and Alessandro Astolfi. "Adaptive control for systems with time-varying parameters." IEEE Transactions on Automatic Control 66.5 (2020): 1986-2001.

---

> > > ### Comment · Reviewer_a2wg · 2024-11-22
> > >
> > > Thanks for the detailed response. My comments are addressed. I am happy the authors dived into adaptive control with time-varying parameter literature. To answer the authors' question (and nothing to change in the paper as such), as far as I am aware, zero parameter identification error is generally not considered possible for arbitrary time-varying parameters; one can only show convergence to a bounded set. However, achieving zero trajectory tracking or state observation error $\tilde{x}$ is possible using additional robust state-feedback terms in the controller/observer development, although there are other disadvantages of using those.

---

> > > > ### Author Response · Authors · 2024-11-22
> > > > **Appreciation for the constructive comments**
> > > >
> > > > Yes, we agree that additional robust terms come at a cost while bringing convergence benefits, such as the requirement for some high-performance actuators.
> > > >
> > > > To sum up, we sincerely appreciate your constructive comments and prompt responses, which helped us improve our paper significantly. Especially, your suggestions have made the theoretical part more rigorous. It is a good revision experience.

---

> > ### Author Response · Authors · 2024-11-24
> > **New theoretical and test results**
> >
> > Dear reviewer a2wg,
> >
> > Under your previous guidance, we have recently worked on integrating the proposed feedback mechanism with adaptive neural ODEs and have achieved some initial results that we would like to share with you.
> >
> > The developed theoretical result with a convergence guarantee is presented in Appendix 8 of the latest manuscript. The adaptive scheme is utilized to update the last layer of the neural ODE, while the feedback scheme is employed to further attenuate learning residuals. We also test this scheme in the quadrotor example, as shown in Figure S14. Note that in the test the feedback gain is set the same as that of the previous feedback neural network. It can seen that the adaptation-enhanced feedback neural network (abbreviated as AdapFNN) achieves performance comparable to the previous feedback neural network, albeit with a slightly larger RMSE.
> >
> > We speculate that the lack of significant performance improvement in the AdapFNN may be due to the final layer of the neural network not being trained analytically, meaning the uncertainty present in the test scenario is not reflected in the last layer. The bilevel training strategy in [1-2] may help improve AdapFNN's performance. We see great potential in exploring this direction further.
> >
> > Thanks for your inspiration!
> >
> > Best Regards,
> >
> > Authors of Submission 8713
> >
> > **Reference**
> >
> > [1] O’Connell, Michael, et al. "Neural-fly enables rapid learning for agile flight in strong winds." Science Robotics 7.66 (2022): eabm6597.
> >
> > [2]Richards, Spencer M., et al. "Control-oriented meta-learning." The International Journal of Robotics Research 42.10 (2023): 777-797.

---

### Official Review · Reviewer_nZLP · 2024-11-04

**Soundness:** 3
**Presentation:** 3
**Contribution:** 3
**Rating:** 8
**Confidence:** 3

**Summary:**

The paper introduces *feedback neural networks*, enhancing the generalization of neural ODEs in continuous-time tasks by adapting learned dynamics through real-time feedback. Inspired by biological systems, the network employs a two-DOF structure with linear feedback (whose convergence is theoretically analyzed) and nonlinear feedback learned via domain randomization. Unlike prior methods, it achieves robust generalization without sacrificing accuracy. Experiments, including trajectory prediction and quadrotor control, demonstrate its improvements over existing methods.

**Strengths:**

1. Clear and promising motivation and concise writing that trace the development of integrating Neural ODEs with feedback structures, progressing from linear to nonlinear feedback.
2. Theoretical analysis is presented for the linear feedback case, demonstrating that prediction error converges to a bounded set with an appropriately chosen feedback gain.
3. Extensive illustrations on simple cases effectively demonstrate their method’s performance compared to standard Neural ODEs.
4. In addition to toy experiments on trajectory prediction, they conduct practical quadrotor tracking experiments, incorporating learned prediction models within an MPC controller, where the proposed FNN-MPC shows superior performance over alternative approaches.

**Weaknesses:**

I have limited experience with Neural ODEs, so I’m not fully confident in assessing the contribution to this area. My main concerns, however, relate to the experimental section. While the current experiments effectively showcase the improvements of FNN, the experimental setup is relatively simple and lacks key baselines in learning dynamics models beyond Neural ODEs, such as [1-3]. Furthermore, it is unclear how flight trajectories were collected and how tracking performance was evaluated. Given the complexity of quadrotor dynamics and aerodynamics, these aspects significantly impact the difficulty of the task and the validity of the results. Although a preliminary overview of quadrotor dynamics is provided in the appendix, additional discussion on environmental conditions, such as wind fields and airflows [4], would help clarify the experimental setting.

[1] Leonard Bauersfeld, et al. NeuroBEM: Hybrid Aerodynamic Quadrotor Model. RSS 2021.

[2] Alessandro Saviolo and Giuseppe Loianno. "Learning quadrotor dynamics for precise, safe, and agile flight control." *Annual Reviews in Control* 55 (2023): 45-60.

[3] Pratyaksh Prabhav Rao et al. "Learning Long-Horizon Predictions for Quadrotor Dynamics". IROS 2024

[4] Leonard Bauersfeld, et al. "Robotics meets Fluid Dynamics: A Characterization of the Induced Airflow below a Quadrotor as a Turbulent Jet".

**Questions:**

See the weaknesses part.

---

> ### Author Response · Authors · 2024-11-19
> **Response to Reviewer nZLP (1/2)**
>
> Thanks a lot for the encouragement and constructive comments from Reviewer nZLP. All your comments have been explicitly considered in the revised manuscript. For your convenience, we have highlighted the modification details both in the following reply and the revised paper.
>
> **Q1: My main concerns, however, relate to the experimental section. While the current experiments effectively showcase the improvements of FNN, the experimental setup is relatively simple and lacks key baselines in learning dynamics models beyond Neural ODEs, such as [1-3].**
>
> **Answer:** Thanks for this valuable comment. Following your advice, we have compared two previous learning-based methods in the quadrotor example. One is the feedforward neural network-based residual model [2] which employs the fully connected neural network to learn the residual model. The other is the adaptive neural network-based residual model [5] in which the last layer of the neural network is regarded as a weighted vector, being adjusted adaptively according to real-time state feedback.  The RMSE of all implemented methods in the quadrotor example are summarized in the following table, where AdapNN-MPC refers to the adaptive neural network method [5] updating the last layer online, and MLP-MPC represents the feedforward neural network-based method [2].
>
> |          | Nomi-MPC | Neural-MPC | FB-MPC | MLP-MPC | AdapNN-MPC | FNN-MPC   |
> | -------- | -------- | ---------- | ------ | ------- | ---------- | --------- |
> | RMSE [m] | 0.248    | 0.167      | 0.203  | 0.182   | 0.151      | **0.093** |
>
> Figure 9 shows the trajectory tracking performance. It can be seen that the performance of MLP-MPC is relatively unsatisfactory compared with that of Neural-MPC. The reason can be attributed to its single-step training manner instead of the multi-step one of the Neural-MPC, leading to a poor multi-step prediction required by MPC. Due to the adaptive ability of the last layer, AdapNN-MPC can handle a certain level of uncertainty. Finally, the proposed FNN-MPC achieves the best tracking performance. The reason can be attributed to the multi-step prediction algorithm of the feedback neural network, which improves the prediction accuracy subject to multiple uncertainties.
>
> **Revisions:** Figure 9 - test results of the quadrotor example; descriptions of compared methods and results in Section 5.2.2 and Appendix A.5.4.

---

> ### Author Response · Authors · 2024-11-19
> **Response to Reviewer nZLP (2/2)**
>
> **Q2: Furthermore, it is unclear how flight trajectories were collected and how tracking performance was evaluated. Given the complexity of quadrotor dynamics and aerodynamics, these aspects significantly impact the difficulty of the task and the validity of the results. Although a preliminary overview of quadrotor dynamics is provided in the appendix, additional discussion on environmental conditions, such as wind fields and airflows [4], would help clarify the experimental setting.**
>
> **Answer:** We appreciate this suggestion. In the revised paper, we have supplemented more details about the way to collect flight trajectories and evaluate tracking performance, Moreover, the environmental condition is introduced further in the Appendix.
>
> 1) When collecting training trajectories, we first randomly sample $3$ positional waypoints in a limited space. Then the minimum snap technique [6] is utilized to optimize a polynomial trajectory connecting these points. Finally the quadrotor with the baseline controller from [7] is commended to follow the planned trajectory. The real flight result is collected as a training trajectory. The above procedure is repeated $40$ times. The collected trajectories are visualized in Figure 8.
> 2) For evaluating the tracking performance of different methods, a Lissajous trajectory out of the training set is chosen as the test case, and the root mean square errors (RMES) between the desired and real flight trajectories with different implemented methods are calculated.
> 3) As for training conditions, we haven't considered the wind disturbance in this work. To approximate a real quadrotor model, the aerodynamic drag is modeled using the liner model from [8-9], in which the coefficients are fitted by real flight data from [7].
>
> **Revisions:**  Figure 8 - training trajectories and the caption; related description about evaluation in section 5.2.1-5.2.2; related description about test conditions in appendix A.5.2.
>
> **Reference:**
>
> [1] Leonard Bauersfeld, et al. "NeuroBEM: Hybrid Aerodynamic Quadrotor Model." *Robotics: Science and Systems.* 2021.
>
> [2] Alessandro Saviolo and Giuseppe Loianno. "Learning quadrotor dynamics for precise, safe, and agile flight control." *Annual Reviews in Control* 55 (2023): 45-60.
>
> [3]  Pratyaksh Prabhav Rao et al. "Learning Long-Horizon Predictions for Quadrotor Dynamics". *International Conference on Intelligent Robots and Systems (IROS)*. 2024.
>
> [4] Bauersfeld, Leonard, et al. "Robotics meets Fluid Dynamics: A Characterization of the Induced Airflow around a Quadrotor." *arXiv preprint arXiv:2403.13321* (2024).
>
> [5] Y. Cheng, et al. "Human motion prediction using semi-adaptable neural networks." *American Control Conference (ACC).* 2019.
>
> [6] D. Mellinger, et al, "Minimum snap trajectory generation and control for quadrotors," *IEEE International Conference on Robotics and Automation (ICRA)*. 2011.
>
> [7] Jia, Jindou, et al. "Accurate high-maneuvering trajectory tracking for quadrotors: A drag utilization method." *IEEE Robotics and Automation Letters* 7.3 (2022): 6966-6973.
>
> [8] Faessler, Matthias, Antonio Franchi, and Davide Scaramuzza. "Differential flatness of quadrotor dynamics subject to rotor drag for accurate tracking of high-speed trajectories." *IEEE Robotics and Automation Letters* 3.2 (2017): 620-626.
>
> [9] Kai, Jean-Marie, et al. "Nonlinear feedback control of quadrotors exploiting first-order drag effects." *IFAC-PapersOnLine* 50.1 (2017): 8189-8195.

---

> ### Author Response · Authors · 2024-11-21
> **Looking forward to your reply**
>
> Dear Reviewer nZLP,
>
> We have provided new experiments and discussions according to your valuable suggestions, which have been absorbed into the revised manuscript. We hope that the new manuscript is made to be stronger with your suggestions.
>
> As the rebuttal deadline is approaching, we sincerely look forward to your reply. Thanks so much for your time and effort!
>
> Best Regards,
>
> Authors of Submission 8713

---

> > ### Comment · Reviewer_nZLP · 2024-11-23
> >
> > Thanks for the response! my questions have been solved, since I am not very familiar with this topic, so I keep my score as it stands now.

---

> > > ### Author Response · Authors · 2024-11-23
> > > **Thank you for the response**
> > >
> > > Dear Reviewer nZLP,
> > >
> > > Thanks for your feedback and dedication to enhancing the quality of our paper. We truly appreciate the valuable suggestions you provided. After rebuttal, we are happy that all the concerns are addressed with two baseline experiments, theoretical guarantees, and several comprehensive ablation studies.
> > >
> > > We are confident that the revised manuscript has been greatly improved and other two reviewers have raised to 8. With all your concerns addressed, we believe our contribution is not a baseline level paper, and hence may we request a score raising? If not, could you please offer us more suggestions so that we can make further improvements to achieve a higher score？
> > >
> > > Thanks a lot for your consideration!
> > >
> > > Best regards,
> > > Authors of Submission 8713

---

> > > > ### Comment · Reviewer_nZLP · 2024-11-25
> > > >
> > > > Increased!

---

> > > > > ### Author Response · Authors · 2024-11-26
> > > > > **Appreciation for the constructive comments**
> > > > >
> > > > > We are truly thankful for your beneficial comments!

---

### Official Review · Reviewer_q2Ng · 2024-11-04

**Soundness:** 3
**Presentation:** 3
**Contribution:** 3
**Rating:** 8
**Confidence:** 3

**Summary:**

This paper studies the challenge of enhancing generalizability and adaptability in Neural ODEs for continuous time-series prediction tasks. It introduces feedback mechanisms into the Neural ODE framework, which dynamically adjust model parameters in response to errors in predictions. The study begins with a theoretical analysis, providing a convergence analysis of linear feedback, and extends to a practical implementation where a neural network learns the feedback mechanism. Experimental results highlight the improved performance of the neural feedback mechanisms over traditional ODE models.

**Strengths:**

1) The paper studies the critical issue of adaptability and generalizability in continuous-time prediction tasks using Neural ODEs, which is essential for real-world applications.
2) By incorporating feedback mechanisms, the approach potentially enhances the adaptability and generalizability of Neural ODEs, making it a contribution to the field.
3) By conducting several experiments, the proposed method demonstrates superior performance across various trajectory and dynamics prediction tasks.

**Weaknesses:**

1) **Comparison with Existing Feedback Mechanisms**: While feedback mechanisms are commonly employed in other neural network-based time-series prediction tasks, this paper does not compare its approach to these established methods. For example, studies such as [1,2] use feedback mechanisms inspired by the Kalman filter and Extended Kalman Filter (EKF) to adapt neural networks for online time-series prediction. Including a comparison with these relevant works would be.

2) **Connection to Continual Learning**: The claims about generalization across new tasks while maintaining performance on previous tasks touch upon the principles of continual learning. A discussion of related works in continual learning, particularly focusing on online continual learning [3] and test-time adaptation [2,4], would provide a richer context.

3) **Evaluation of Theoretical Claims and Linear Feedback**: The paper presents a theorem on the convergence of linear feedback to a bounded error but does not empirically evaluate this claim. Designing a simple toy experiment to test this theorem and discussing the practical implications of the assumptions would enhance the credibility and thoroughness of the research. In addition, reporting the results of Linear feedback on several trajectory prediction tasks in the experiment will improve the quality of the paper.

4) There are minor typos and notation errors that need correction for clarity. Line 167: $\dot{z}=[\hat{f}^T,\hat{f}^T]^\top$ -> $\dot{z}=[f^T,\hat{f}^T]^\top$ ; Line 257：  'sate'  ->  'state'.

[1] Y. Cheng, et al. "Human motion prediction using semi-adaptable neural networks." 2019 American Control Conference (ACC). IEEE, 2019.
[2] A. Abuduweili,  et al. "Robust online model adaptation by extended kalman filter with exponential moving average and dynamic multi-epoch strategy." Learning for Dynamics and Control, 2020.
[3] Y. Ghunaim, et al. "Real-time evaluation in online continual learning: A new hope." Proceedings of the IEEE/CVF conference on computer vision and pattern recognition. 2023.
[4] J. Liang, et al. "A comprehensive survey on test-time adaptation under distribution shifts." International Journal of Computer Vision (2024): 1-34.

**Questions:**

1) Review Related Work on Feedback Adaptation in Time-Series Prediction [1,2].
2) Explore the Relationship with Continual Learning [2,3,4].
3) Perform Experiments on Linear Feedback.

---

> ### Author Response · Authors · 2024-11-19
> **Response to Reviewer q2Ng (1/3)**
>
> We sincerely thank the reviewer q2Ng for the insightful and constructive comments. We are glad that the reviewer acknowledges that the research topic is important and the proposed method is effective on all considered tasks. Here we answer all the questions and provide suggested compared experiments. We hope the responses can address the concerns. In addition, we have submitted a revised manuscript where we mark all the suggested figures and analytics in magenta color.
>
> **Q1: Comparison with Existing Feedback Mechanisms: While feedback mechanisms are commonly employed in other neural network-based time-series prediction tasks, this paper does not compare its approach to these established methods. For example, studies such as [1,2] use feedback mechanisms inspired by the Kalman filter and Extended Kalman Filter (EKF) to adapt neural networks for online time-series prediction.**
>
> **Answer:** Thanks for this valuable comment. Two baseline comparisons have been supplemented in the revised manuscript. We also noticed previous feedback-based neural network methods [1-2, 5-6]. Especially in [1, 5-6], the dynamical uncertainty is learned by the neural network, in which the last layer of the network is regarded as a weighted vector, being adjusted adaptively according to real-time state feedback. In such a paragram, the effect of online time-vary uncertainty is attributed to the last layer. Differently, we try to correct the latent dynamics directly within the framework of neural ODEs, which are trained on trajectory instead of labels of state derivative. We think both have their advantages. The proposed method is more explainable benefiting from the mechanism of correcting latent dynamics, and feedback can be learned by a neural form.
>
> By following your advice, we have compared the previous feedback-based neural network method [1] in the quadrotor example. Moreover, a learning-based model [7] that employs the fully connected neural network to learn aerodynamic drag is also compared. The RMSE of all implemented methods in the quadrotor example are summarized in the following table, where AdapNN-MPC refers to the feedback-based neural network method [1] updating the last layer adaptively, and MLP-MPC represents the added learning-based method [7].
>
> |          | Nomi-MPC | Neural-MPC | FB-MPC | MLP-MPC | AdapNN-MPC | FNN-MPC   |
> | -------- | -------- | ---------- | ------ | ------- | ---------- | --------- |
> | RMSE [m] | 0.248    | 0.167      | 0.203  | 0.182   | 0.151      | **0.093** |
>
> Figure 9 shows the trajectory tracking performance. It can be seen that the performance of MLP-MPC is relatively unsatisfactory compared with that of Neural-MPC. The reason can be attributed to its single-step training manner instead of the multi-step one of the Neural-MPC, leading to a poor multi-step prediction required by MPC. Due to the adaptive ability of the last layer, AdapNN-MPC can handle a certain level of uncertainty. Finally, FNN-MPC achieves the best tracking performance. The reason can be attributed to the multi-step prediction algorithm of the feedback neural network, which improves the prediction accuracy subject to multiple uncertainties.
>
> **Revisions:** Figure 9 - test results of the quadrotor example; descriptions of compared methods and results in Section 5.2.2 and Appendix A.5.4.

---

> ### Author Response · Authors · 2024-11-19
> **Response to Reviewer q2Ng (2/3)**
>
> **Q2: Connection to Continual Learning: The claims about generalization across new tasks while maintaining performance on previous tasks touch upon the principles of continual learning. A discussion of related works in continual learning, particularly focusing on online continual learning [3] and test-time adaptation [2,4], would provide a richer context.**
>
> **Answer:** By following your advice, we have further enriched the related work to cover online continual learning [3] and test-time adaptation [1-2,4]. More related works like [5-6] all also included. For easy viewing, we rephrase the added part here.
>
> Recently, online continual learning [3] and test-time adaption [4] have emerged as promising solutions to handle unknown test distribution shifts. Online continual learning focuses on the reduction of real-time training load, aiming at generalizing across new tasks while maintaining performance on previous tasks. Test-time adaption tries to utilize real-time unlabeled data to obtain self-adapted models. For example, an extended *kalman* filter-based adaption algorithm with a forgetting factor is developed by [2] to generalize neural network-based models. Moreover, in order to improve the flexibility of neural networks, the last layer of networks can be regarded as a weighted vector, which can be adjusted adaptively according to real-time state feedback [1, 5-6]. The training for separating the last layer and front structure can be carried out within a bi-level optimization framework. In such a paradigm, the uncertainty out of training sets is reflected on the last layer of networks, which can be online adjusted in a control-oriented [6] or regression-oriented [5] fashion. [8] further develops real-time weight adaptation laws for all layers of feedforward neural networks, with stability guarantees.
>
> Different from the above retraining or adaption strategy, the presented method directly corrects the learned latent dynamics of neural ODEs with real-time feedback, yielding a two-DOF network structure. Moreover, the feedback can be learned in a neural form. Integrating adaptive neural ODEs with the developed feedback mechanism may be a valuable research direction.
>
> **Revision:** Section 6.3 - Real-time retraining and adaption.
>
> **Q3: Evaluation of Theoretical Claims and Linear Feedback: The paper presents a theorem on the convergence of linear feedback to a bounded error but does not empirically evaluate this claim. Designing a simple toy experiment to test this theorem and discussing the practical implications of the assumptions would enhance the credibility and thoroughness of the research. In addition, reporting the results of Linear feedback on several trajectory prediction tasks in the experiment will improve the quality of the paper.**
>
> **Answer: **The performance of developed linear feedback was evaluated in Figure 5. In this revision, we have improved the caption of Figure 5 to highlight its implemented condition.  Moreover, we further added Figure S1 to show the prediction error satisfies the theoretical bound of Theorem 1.
>
> The assumption can also cover common step disturbances. We also added related test results on common step disturbances in Figure S12. As for unbounded learning residuals violating the assumption, we think it is still a major challenge in learning fields. It reveals that neural networks have completely lost the representational ability to target uncertainties. The best strategy may be retraining the networks based on fresh datasets, like an online continual learning mission [3].  The above discussion has been provided in Appendix A.2.1. Future work will keep this limitation in mind and try to follow the route of online continual learning to address it.
>
> We also conducted the ablation study on the feedback gain. Theorem 1 shows the converged bound of ${{\tilde x}}(t)$ relies on the norm bound $\gamma$ in Assumption 1 and the feedback gain $L$. The converged bound increases with the increase of $\gamma$, and decreases with the increase of $\lambda_m({L})$. However, the amplitude of $\lambda_m({L})$ is limited since the feedback ${x}$ is usually noised. This analysis was tested in the spiral curve example, as shown in Figure 4. Following your advice, the related statements have been also improved in the revised manuscript. As for irregular object and quadrotor tests, we also used linear feedback due to its practicability.
>
> **Revisions:** The caption of Figure 5;  Figure S1 - test of convergence set; Figure S12 - test on step disturbances; discussion of Assumption 1 in Appendix A.2.1; the caption of Figure 4; related description about Figure 4 in section 3.4.

---

> ### Author Response · Authors · 2024-11-19
> **Response to Reviewer q2Ng (3/3)**
>
> **Q4: There are minor typos and notation errors that need correction for clarity. Line 167: $\dot{z} = \left[\hat{f}^T,\hat{f}^T\right]^T \rightarrow \dot{z} = \left[{f}^T,\hat{f}^T\right]^T$;  Line 257： 'sate' -> 'state'.**
>
> **Answer:** Thanks for carefully checking this manuscript. These minor issues have been corrected.
>
> **Revisions:**  Lines 167 and 257.
>
> **Reference:**
>
> [1] Y. Cheng, et al. "Human motion prediction using semi-adaptable neural networks." *American Control Conference (ACC)*. 2019.
>
> [2] A. Abuduweili, et al. "Robust online model adaptation by extended kalman filter with exponential moving average and dynamic multi-epoch strategy." *Learning for Dynamics and Control*. 2020.
>
> [3] Y. Ghunaim, et al. "Real-time evaluation in online continual learning: A new hope." *Proceedings of the IEEE/CVF Conference on Computer Vision and Pattern Recognition*. 2023.
>
> [4] J. Liang, et al. "A comprehensive survey on test-time adaptation under distribution shifts." *International Journal of Computer Vision* (2024): 1-34.
>
> [5] O’Connell, Michael, et al. "Neural-fly enables rapid learning for agile flight in strong winds." *Science Robotics* 7.66 (2022): eabm6597.
>
> [6]Richards, Spencer M., et al. "Control-oriented meta-learning." *The International Journal of Robotics Research* 42.10 (2023): 777-797.
>
> [7] Alessandro Saviolo and Giuseppe Loianno. "Learning quadrotor dynamics for precise, safe, and agile flight control." *Annual Reviews in Control* 55 (2023): 45-60.
>
> [8] Patil, Omkar Sudhir, et al. "Lyapunov-derived control and adaptive update laws for inner and outer layer weights of a deep neural network." *IEEE Control Systems Letters* 6 (2021): 1855-1860.

---

> ### Author Response · Authors · 2024-11-21
> **Looking forward to your reply**
>
> Dear Reviewer q2Ng,
>
> We have provided new experiments and discussions according to your valuable suggestions, which have been absorbed into the revised manuscript. We hope that the new manuscript is made to be stronger with your suggestions.
>
> As the rebuttal deadline is approaching, we sincerely look forward to your reply. Thanks so much for your time and effort!
>
> Best Regards,
>
> Authors of Submission 8713

---

> ### Author Response · Authors · 2024-11-24
> **New theoretical and test results**
>
> Dear Reviewer q2Ng,
>
> We have recently integrated the proposed feedback mechanism with the adaptive neural network [1] you mentioned, leading to new theoretical and experimental results. For further details, please refer to Appendix 8 of our latest manuscript.
>
> As the rebuttal deadline approaches, we eagerly await your response.
>
> Best regards,
>
> Authors of Submission 8713
>
> **Reference:**
>
> [1] Y. Cheng, et al. "Human motion prediction using semi-adaptable neural networks." American Control Conference (ACC). 2019.

---

> > ### Comment · Reviewer_q2Ng · 2024-11-25
> > **Post rebuttal comment**
> >
> > Thank you for addressing my concerns in your response. The authors have resolved most of the issues I raised. I consider this to be a good contribution. Accordingly, I am raising my score from 6 to 8.

---

> > > ### Author Response · Authors · 2024-11-26
> > > **Appreciation for the constructive comments**
> > >
> > > We sincerely appreciate your helpful suggestions! Thanks for your time and effort!

---

### Author Response · Authors · 2024-11-19
**Summary of the rebuttal and the major changes of revised manuscript**

Dear reviewers,

We would like to express our heartfelt gratitude for your invaluable time and meticulous attention in reviewing our manuscript. We appreciate that all four reviewers gave us positive feedback with insightful suggestions that can help us improve the quality and rigor of our work.

We appreciate that the reviewers acknowledge the advantages of our work: **“This paper makes a strong contribution. The paper is well-written and easy to follow. The technical development with the use of closed-loop state error feedback term is well-motivated. I found the paper an interesting read.”** (reviewer a2wg), **“The paper studies the critical issue of adaptability and generalizability, which is essential for real-world applications.”** (reviewer q2Ng), **“Clear and promising motivation and concise writing.”** (reviewer nZLP), **“This design demonstrates innovation and practical value in current research. This research has significant implications for neural network architecture design and practical engineering applications, showing broad application potential.”** (reviewer Lpz2).

On the other hand, we have diligently addressed all the issues by conducting extensive experiments to support our responses. A revised manuscript has been submitted, accompanied by an appendix that delineates all revisions, highlighted in **magenta color** for clarity. Owing to space constraints, selected experiments have been incorporated into the main manuscript while supplementary experiments have been included in the appendix. Allow me to summarize the significant alterations made in both the rebuttal and the revised manuscript:

1. **Expanded Baselines**: Included two additional methods (feedforward neural network and adaptive neural network-based residual learning) as baselines to demonstrate the effectiveness of the proposed method in the quadrotor test (Section 5.2.2, Figure 9, Appendix A.5.4).
2. **Theoretical consolidation:** Improved the proof of Theorem 1, leading to a tighter convergence bound (Appendix A.2.1) and a discrete-time stability analysis (Appendix A.2.2).
3. **Ablation Study Enrichment**: Expanded several ablation studies on linear feedback unit (Appendix A.6.1), neural/nonlinear feedback unit (Appendix A.6.2), and decay rate (Appendix A.6.3), offering insights into each part of the feedback neural network.
4. **Related Work Enrichment:** Discussed more previous research about online continual learning and test-time adaptation (Section 6.3),  highlighting the method's unique feedback mechanism.
5. **Test Details Enrichment:** Provided more details about the way to collect flight trajectories (Figure 8), evaluate tracking performance (Section 5.2.1), and model aerodynamic drag (Appendix A.5.2), offering additional insights into the test condition.
6. **Writing Revision**:  Revised all notation definitions and minor mathematical issues.
7. **Others:** Provided a test to verify the theoretical bounded set (Figure S1),  a test on sudden disturbance (Figure S12), a discussion on training cost (Appendix A.7), and a limitation on manually tunning feedback gain and decay rate (Limitation).

Best Regards,

Authors of Submission 8713

---

### Meta-Review · Area_Chair_5tFL · 2024-12-26

**Metareview:**

The paper introduces feedback neural networks to enhance the generalization capabilities of Neural ODEs for continuous-time prediction tasks. The key innovation is a two-degree-of-freedom (Two-DOF) architecture that combines linear and nonlinear feedback mechanisms to correct prediction errors in learned latent dynamics.

positive:

- combining feedback with Neural ODEs to improve generalization without sacrificing accuracy (not new, but valuable to see the results).
- sound theoretical foundation with convergence analysis for the linear feedback case
- experimental validation in real-world applications like quadrotor control
- the feedback mechanism can be integrated into existing trained Neural ODEs without requiring retraining

Weaknesses:

- somewhat idealized assumptions in the convergence analysis (bounded learning residuals)
- manual tuning required for feedback gain matrix
- could benefit from more extensive comparison with state-of-the-art baselines

Authors managed to address most of reviewers concerns and acknowledged the limitations of their approach. I vote for accepting the paper.

**Additional Comments On Reviewer Discussion:**

Here is what the authors did:

- they added two new baselines (feedforward and adaptive neural networks)
- they authors provided additional proofs, including discrete-time stability analysis
- they added extensive ablation studies on linear/nonlinear feedback components and hyperparameters
- they improved notation definitions and theoretical bounds and added more information about data collection, training procedures, and computational costs.

All reviewers were satisfied with the authors' responses.

---

### Decision · Program_Chairs · 2025-01-22

Accept (Oral)